# WSVD: Weighted Low-Rank Approximation for Fast and Efficient Execution of Low-Precision Vision-Language Models

**Haiyu Wang**[1]  **Yutong Wang**[1]  **Jack Jiang**[2]  **Sai Qian Zhang**[1,2]
[1]Tandon School of Engineering, New York University
[2]Courant Institute of Mathematical Sciences, New York University
{hw3689,yw6594,jj2513,sai.zhang}@nyu.edu

## Abstract

Singular Value Decomposition (SVD) has become an important technique for reducing the computational burden of Vision Language Models (VLMs), which play a central role in tasks such as image captioning and visual question answering. Although multiple prior works have proposed efficient SVD variants to enable low-rank operations, we find that in practice it remains difficult to achieve substantial latency reduction during model execution. To address this limitation, we introduce a new computational pattern and apply SVD at a finer granularity, enabling real and measurable improvements in execution latency. Furthermore, recognizing that weight elements differ in their relative importance, we adaptively allocate relative importance to each element during SVD process to better preserve accuracy, then extend this framework with quantization applied to both weights and activations, resulting in a highly efficient VLM. Collectively, we introduce *Weighted SVD* (WSVD), which outperforms other approaches by achieving over $1.8\times$ decoding speedup while preserving accuracy. We open source our code at: `https://github.com/SAI-Lab-NYU/WSVD`.

## 1 Introduction

Vision–language models (VLMs) represent a key frontier in artificial intelligence, as they connect visual recognition with natural language comprehension. By jointly processing imagery and text, these models enable diverse applications, including automatic image description (Zhou et al., 2020; Hu et al., 2022; Chen et al., 2022; Dzabraev et al., 2024), visual question answering (Chappuis et al., 2022; Bazi et al., 2023; Wang et al., 2024b), and semantic search over multimodal data (Li et al., 2024b; Sun et al., 2025). However, the impressive capabilities of VLMs come at the expense of significant resource demands. The joint encoding of large-scale visual and linguistic inputs requires heavy computation, and the autoregressive generation of tokens further stresses memory bandwidth, introducing major inference bottlenecks.

To reduce the computational cost of large models, low-rank decomposition has recently attracted increasing attention (Wang et al., 2025c; Yuan et al., 2023b; Wang et al., 2024d; Li et al., 2025; 2024c; Chang et al., 2024; Wang et al., 2025a). By factorizing the query (Q), key (K), and value (V) matrices within self-attention blocks into low-rank components, prior work has shown significant reductions in computational complexity and weight storage, thereby improving efficiency. However, based on our practical system-level implementation, we observe that applying SVD-based decomposition to the QKV matrices does not consistently yield latency improvements; in fact, it can sometimes incur even higher computational costs for some VLMs.

To investigate this, we first evaluate the latency of VLMs and find that the root cause lies in the recomputation of the KV vectors introduced by low-rank factorization, which requires multiple rounds of memory access to the latent data and ultimately increases memory traffic. To overcome this limitation, we propose a new computational pattern that applies SVD at a finer granularity, thereby achieving tangible and measurable improvements in execution latency.

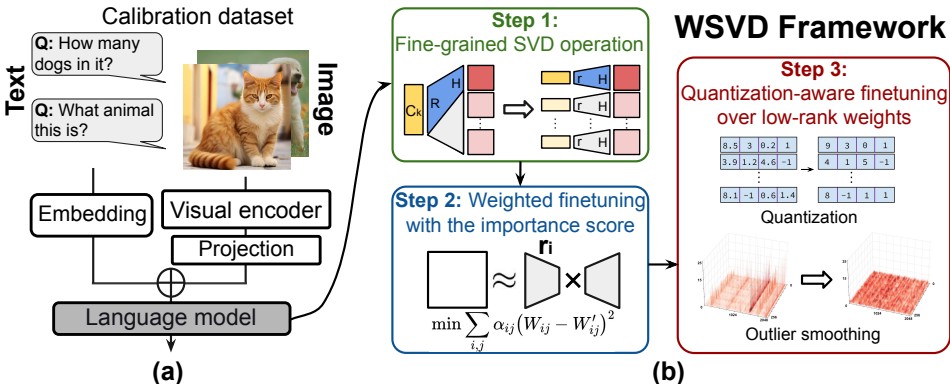

Figure 1: (a) Architecture of vision-language model. (b) Overview of WSVD framework.

Furthermore, building on prior work (Yu et al., 2024b) demonstrating that certain weight elements play a critical role in VLM accuracy, we note that standard SVD operations treat all weights equally when truncating them for low-rank approximation. To address this, we adaptively allocate relative importance for each weight element during SVD to better preserve performance. To further enhance computational efficiency, we apply low-precision quantization to the low-rank VLM and finetune it to mitigate accuracy loss. Collectively, these steps yield a low-precision, low-rank VLM with exceptionally low execution latency. Our contributions are summarized as follows:

- The WSVD scheme applies SVD separately to each attention head, fundamentally reducing memory access and computational cost during the decoding stage, and resulting in significantly lower VLM execution latency compared to prior solutions.

- To mitigate the accuracy drop introduced by the per-head SVD scheme, WSVD incorporates local weighted finetuning, where an importance score is assigned to each weight element during the SVD stage. This weighted decomposition produces low-rank weight matrices with minimal impact on VLM accuracy.

- WSVD applies quantization alongside SVD decomposition to both the weights and activations of the VLM. To further enhance efficiency, it incorporates an outlier elimination strategy within the SVD framework and locally finetunes the decomposed matrices, achieving improved accuracy while substantially reducing memory and computational cost.

## 2 RELATED WORK

### 2.1 VISION LANGUAGE MODEL

Vision–Language Models (VLMs) (Li et al., 2022; 2023; Liu et al., 2023; Dai et al., 2023; Beyer et al., 2024; Grattafiori et al., 2024; Wang et al., 2024c) build on the foundation of Large Language Models (LLMs) by incorporating visual signals in addition to textual input, thereby enabling multimodal tasks such as image captioning and visual question answering (VQA). Representative systems like BLIP and InstructBLIP (Li et al., 2022; 2023) leverage large-scale data curation and visual instruction tuning to better align their responses with human intent, particularly under zero-shot evaluation. A widely adopted framework, shown in Figure 1 (a), encodes images into visual tokens via a vision backbone, concatenates them with text tokens, and feeds the combined sequence into a language model for output generation. This simple yet effective concatenation strategy underpins popular VLMs such as the LLaVA family (Liu et al., 2023), SmolVLM (Marafioti et al., 2025), PaLI-Gemma (Beyer et al., 2024), and Qwen-VL (Wang et al., 2024c). Despite their strong performance, these models are often computationally heavy and difficult to deploy on devices with limited resources. To address this, compact designs have been introduced. TinyGPT-V (Yuan et al., 2023a) and TinyLLaVA (Zhou et al., 2024) pursue scaled-down yet efficient alternatives, while SmolVLM (Marafioti et al., 2025) presents a family of lightweight models with one to three billion parameters that preserve competitive accuracy while significantly lowering hardware requirements.

## 2.2 SINGULAR VALUE DECOMPOSITION FOR LARGE MODELS

Singular Value Decomposition (SVD) (Jolliffe & Cadima, 2016) is a fundamental tool in matrix factorization that represents a matrix $W \in \mathbb{R}^{m \times n}$ as $W = U\Sigma V^T$, where $U$ and $V$ are orthogonal matrices containing the left and right singular vectors, and $\Sigma$ is a diagonal matrix with non-negative singular values sorted in descending order. By retaining only the leading $r$ singular values and their associated vectors, one obtains a compact rank-$r$ approximation:

$$W \approx U_r \Sigma_r V_r^T \tag{1}$$

with $U_r \in \mathbb{R}^{m \times r}$, $\Sigma_r \in \mathbb{R}^{r \times r}$, and $V_r \in \mathbb{R}^{n \times r}$. This form can equivalently be written as $W \approx AB$, where $A = U_r \Sigma_r^{1/2}$ and $B = \Sigma_r^{1/2} V_r^T$. Such low-rank approximations capture the dominant structure of $W$, allowing dimensionality reduction, compression, and faster computation. SVD has been extensively studied as a compression strategy for LLMs (Wang et al., 2025c; Yuan et al., 2023b; Wang et al., 2024d; Li et al., 2025; 2024c; Chang et al., 2024; Wang et al., 2025a). Early work (Noach & Goldberg, 2020) applied vanilla SVD directly to weight matrices, but the method suffered from considerable approximation errors. Subsequent techniques refined this approach: FWSVD (Hsu et al., 2022) incorporates Fisher information (Ly et al., 2017) to rank parameter importance, ASVD (Yuan et al., 2023b) accounts for activation outliers, and SVD-LLM (Wang et al., 2024d) explicitly minimizes the loss introduced by discarded singular values.

While most efforts have focused on compressing model weights, it can also be used for KV cache compression (Chang et al., 2024; Yu et al., 2024a). In particular, the key and value projection matrices can be factorized as $W_K = A_K B_K$ and $W_V = A_V B_V$. For a given input $X$, this allows the KV cache to store only the low-dimensional latent vectors $C_K = X A_K$ and $C_V = X A_V$, thereby reducing cache size. During decoding, the original KV representations are reconstructed via $K = C_K B_K$ and $V = C_V B_V$. More recent innovations include AdaSVD (Li et al., 2025), which dynamically adjusts compression rates based on per-layer sensitivity, and SVD-LLM2 (Wang et al., 2025c), which optimizes truncation strategies using theoretical error analysis.

## 2.3 FISHER-BASED IMPORTANCE AND WEIGHTED MATRIX FACTORIZATION

Fisher information has been widely used as a measure of parameter importance in continual learning (Kirkpatrick et al., 2017) and in pruning and compression (Liu et al., 2021; Singh & Alistarh, 2020). Weighted low-rank approximation has been explored in matrix completion and recommendation, where each entry carries a confidence weight (Srebro & Jaakkola, 2003). More recently, FWSVD (Hsu et al., 2022) incorporates Fisher information into low-rank factorization by assigning a single Fisher-based weight to each row and applying SVD to a pre-scaled matrix, yielding a coarse row-wise weighting. On the interpretability side, gradient-based attribution and layer-wise relevance propagation methods (Ancona et al., 2017; Bach et al., 2015) also use importance weights, but primarily for explanation rather than compression. In contrast, WSVD uses element-wise Fisher weights to directly guide both local fine-tuning and quantization-aware training.

## 2.4 FLASH DECODING

FlashAttention (Dao et al., 2022) is an IO-aware attention algorithm that leverages tiling and kernel fusion to reduce memory traffic and improve GPU utilization. By keeping query, key, and value tiles in on-chip memory and streaming them through a fused kernel, FlashAttention avoids materializing large intermediate attention matrices, thereby lowering memory footprint and achieving substantial speedups in training and inference.

Building on this idea, Flash Decoding (Dao et al., 2023) extends FlashAttention to the autoregressive decoding setting. Instead of materializing and reloading the entire KV cache for each step, it streams $K$ and $V$ in sequence tiles and incrementally updates online softmax statistics. This block-wise processing exposes additional parallelism along the sequence length dimension, complementing the existing head- and batch-level parallelism in FlashAttention, and thereby improves GPU utilization. As a result, Flash Decoding achieves both lower memory traffic and higher throughput, and has become the de facto baseline for efficient inference in large language and vision-language models. Our WSVD system further builds on Flash Decoding by integrating low-rank reconstruction into the fused kernel pipeline, reducing memory overhead while preserving its efficiency (see Section 3.4).

Figure 2: (a) Latency evaluation of VLM including self-attention (SA) and feed-forward (FFN) modules. (b) Conventional SVD: the left side illustrates SVD of $W_k$, and the right side shows the reconstruction of $K_h$ from the shared latent. (c) Per-head SVD: the left side illustrates per-head SVD of $W_{Kh}$, and right side shows per-head reconstruction of $K_h$ from per-head latent.

## 3 METHOD

An overview of WSVD is presented in Figure 1 (b), which consists of three key components: (i) Per-head SVD operations for reduced latency (Section 3.1), (ii) WSVD with dynamic importance scoring (Section 3.2), and (iii) quantization-aware finetuning for low-rank VLMs (Section 3.3).

### 3.1 FINE-GRAINED PER-HEAD SVD OPERATION FOR REDUCED LATENCY

Prior studies have shown that VLM decoding is predominantly memory-bound, as long image-token sequences enlarge the KV cache and each generated token requires accessing the large KV cache, with overall latency bottlenecked by memory access. As discussed in Section 2.2, conventional SVD-based approaches (Chang et al., 2024; Wang et al., 2025d; 2024d) address this by decomposing projection matrices (e.g., $Q$, $K$, and $V$), thereby reducing parameter count and storing low-rank latent representations $C_K$ and $C_V$. This strategy not only decreases computation and runtime in the prefill stage but also reduces cache size, offering potential I/O savings during the decoding stage.

However, in practice, we find that reconstructing $K$ and $V$ from low-rank latents introduces substantial overhead, leading to even higher decoding latency than the original uncompressed model. Specifically, we profile the single-layer decoding latency of LLaVA-Next 7B (Zhou et al., 2024) on an RTX 4090, comparing standard flash decoding without SVD against an SVD baseline that caches low-rank latents. In this baseline, both the rank ratio and cache size are reduced to 50% as before. With a batch size of 16 and a KV cache length of 8192, the results (Figure 2 (a)) show that SVD scheme incurs substantially higher latency compared to flash decoding.

To pinpoint the cause of this latency growth, we observe that the overhead arises from decomposing the entire $K$ and $V$ matrices. Taking $K$ as an example, after SVD we obtain $W_K = A_K B_K$, where $A_K \in \mathbb{R}^{E \times R}$ and $B_K \in \mathbb{R}^{R \times E}$, with $E$ denoting the embedding dimension and $R$ the truncated rank. For each head $h$, the key projection is computed as $W_{Kh} = A_K B_{Kh}$, where $B_{Kh} \in \mathbb{R}^{R \times H}$ and $H$ is the head dimension (Figure 2(b)). During inference, the latent representation $C_K = X A_K \in \mathbb{R}^{L \times R}$ is cached across sequence length $L$, and each head's key vector is reconstructed as $K_h = C_K B_{Kh}$. This reconstruction introduces a computational cost of $\gamma_{\text{svd}} = LRH$ per head. Compared with directly storing the $K$ matrix of size $LE$, although caching $C_K$ reduces storage to $LR$, **reconstructing $W_{Kh}$ for each head requires accessing the entire $C_K$**, which has a size $LR$. As a result, the effective memory footprint becomes $\eta_{\text{svd}} = LR$ per head, thereby increasing decoding latency. Similar argument holds trues for the computation of value vector $V$.

To mitigate this overhead, our WSVD approach applies SVD directly to the submatrices of $W_K$ and $W_V$ corresponding to each head, rather than decomposing the entire matrices, as illustrated in Figure 2 (c). Specifically, for head $h$, the submatrix $W_{Kh} \in \mathbb{R}^{E \times H}$ is decomposed as $W_{Kh} = A_{Kh} B_{Kh}$, where $A_{Kh} \in \mathbb{R}^{E \times r}$ and $B_{Kh} \in \mathbb{R}^{r \times H}$. The rank $r$ is obtained by truncating the $H$ singular values of $W_{Kh}$. Since $H \ll E$, the per-head rank $r$ is typically much smaller than $R$. For each head $h$, the latent representation is computed as $C_{Kh} = X A_{Kh} \in \mathbb{R}^{L \times r}$ and stored in the cache. During decoding, the corresponding key vector is reconstructed as $K_h = C_{Kh} B_{Kh}$. Unlike the conventional SVD approach shown in Figure 2 (b), this design eliminates the need to repeatedly load a large shared latent representation $C_K$, since **each head can be reconstructed directly from its own latent** $C_{Kh}$. With this design, the memory footprint is reduced to $\eta_{\text{wsvd}} = Lr$, since only the latent vector $C_{Kh}$ needs to be stored, and the computational cost of reconstructing $K_h$ becomes $\gamma_{\text{wsvd}} = LrH$, where $r \ll R$. A similar computation applies to the reconstruction of $V$.

To evaluate the saving analytically, the per-head SVD scheme shown in Figure 2 (c) reduces both memory traffic and computational cost, thereby enabling practical decoding acceleration, as demonstrated in Section 4.4. In particular,

$$\frac{\gamma_{\text{wsvd}}}{\gamma_{\text{svd}}} = \frac{\eta_{\text{wsvd}}}{\eta_{\text{svd}}} = \frac{r}{R}, \quad r \ll R. \tag{2}$$

Thus, both computational cost and memory footprint for latent storage are reduced by a factor of $r/R$. Compared to the original SVD-based scheme, WSVD further reduces the weight parameter count from $\alpha_{\text{orig}} = EH$ per head to $\alpha_{\text{wsvd}} = Er + rH$, and lowers the KV-cache size from $\eta_{\text{orig}} = LH$ to $\eta_{\text{wsvd}} = Lr$. These improvements are quantified by the parameter size ratio $\rho_1$ and the cache size ratio $\rho_2$ for KV vector storage.

$$\rho_1 = \frac{\alpha_{\text{wsvd}}}{\alpha_{\text{orig}}} = \frac{(E + H) \times r}{E \times H} = (1 + \frac{H}{E}) \cdot \frac{r}{H}, \qquad \rho_2 = \frac{\eta_{\text{wsvd}}}{\eta_{\text{orig}}} = \frac{r}{H}. \tag{3}$$

However, the per-head SVD in the WSVD scheme also amplifies approximation errors, making accuracy degradation harder to control compared to conventional SVD applied to the full $W_k$. Next, we describe a local weighted finetuning scheme to mitigate the accuracy drop.

## 3.2 SVD with Local Weighted Finetuning

Conventional SVD converts a full-rank input matrix into a low-rank representation, but one limitation is that it cannot control the relative contribution of different weights after decomposition. Prior work (Yu et al., 2024b) has shown that in large models, weights vary significantly in their importance to final accuracy. In particular, some "superweights" are highly sensitive, where even small changes in magnitude can cause a substantial drop in accuracy. Therefore, it is crucial to incorporate this notion of importance when performing SVD, resulting in a weighted low-rank decomposition.

The first question is how to evaluate the importance of a weight element. To formalize this, let $\mathcal{D}$ denote the data distribution over calibration samples $x$, and let $\ell(W; x)$ denote the training loss of sample $x$. The importance score of each element in $W_K$ with respect to final accuracy can be estimated as:

$$G_K = \mathbb{E}_{x \sim \mathcal{D}}\big[\nabla_{W_K} \ell(W; x)\big]. \tag{4}$$

A weight entry with a large gradient magnitude indicates that even a small change in this element has a substantial effect on the expected model loss. Accordingly, $G_K$ can be interpreted as an importance score that links parameter updates to their impact on performance.

This estimation of training loss impact can be refined using the Fisher Information Matrix (FIM), which quantifies parameter importance as the expected sensitivity of the log-likelihood with respect to model parameters. A second-order Taylor expansion of the expected loss around the current parameter values yields:

$$\Delta \mathcal{L} = \mathbb{E}_{x \sim \mathcal{D}}\big[\ell(W + \Delta W; x) - \ell(W; x)\big] \tag{5}$$

$$\approx \tfrac{1}{2} \Delta W^\top \Big( \mathbb{E}_x\big[\nabla_W^2 \ell(W; x)\big]\Big) \Delta W. \tag{6}$$

To make the computation of the Hessian tractable, it can be approximated by a diagonal matrix, where each diagonal entry corresponds to the Fisher importance score of the parameter. For example, the vector of Fisher information score $F_K$ for $W_K$ can be computed as:

$$F_K = \mathbb{E}_{x \sim \mathcal{D}}\big[g_K(x) \odot g_K(x)\big], \quad g_K(x) = \nabla_{W_K} \ell(W; x) \tag{7}$$

where $\odot$ denotes elementwise multiplication. Motivated by these observations, we propose a weighted local fine-tuning mechanism that performs SVD while incorporating the relative importance of each weight element, quantified by its Fisher information score. Specifically, the objective function can be described as:

$$\min_{A_K, B_K} \big\| F_K^{1/2} \odot (W_K - A_K B_K) \big\|_F^2 \tag{8}$$

where $A_K$, $B_K$ are the low-rank decomposition to estimate $W_K$. In the context of per-head SVD described in Section 3.1, this optimization can be applied across the SVD for the weight matrices for each head $h$, and the objective function can be depicted as:

$$\min_{A_{Kh}, B_{Kh}} \sum_h \big\| F_{Kh}^{1/2} \odot (W_{Kh} - A_{Kh} B_{Kh}) \big\|_F^2 \tag{9}$$

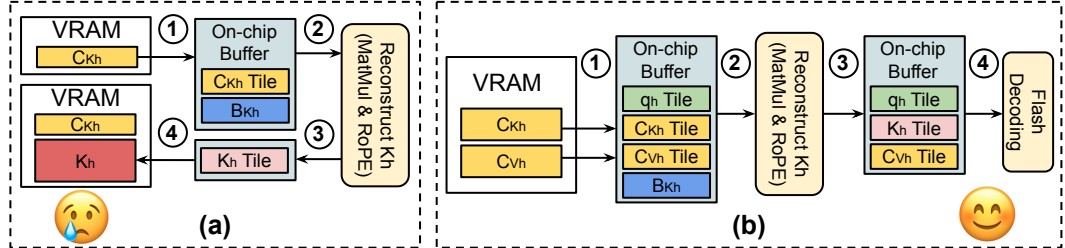

Figure 3: (a) Naive reconstruction requires materializing and writing back full $K_h$ to VRAM (global GPU memory), leading to excessive memory usage and I/O. (b) Our fused kernel consumes $C_{Kh}$ and $C_{Vh}$ tiles on-chip with flash decoding, reducing both peak memory footprint and I/O traffic. All the step numbers are shown in circle.

where $A_{Kh}$ and $B_{Kh}$ denote the low-rank approximation of $W_{Kh}$. Since no analytical solution exists for this problem, it is solved by fine-tuning $A_{Kh}$ and $B_{Kh}$ until convergence. The same loss formulation can be applied to other projection matrices in the model (e.g., $W_Q, W_V$, or feed-forward layers), providing a general framework for gradient-weighted fine-tuning after SVD truncation. The WSVD procedure is summarized in Algorithm 1.

### 3.3 LOCAL QUANTIZATION-AWARE TRAINING FOR LOW-PRECISION WSVD

To further reduce model size and cache footprint, we apply low-precision quantization to the low-rank model parameters and the input and mitigate accuracy loss using local quantization-aware training (QAT). To address channel-wise outliers in the input $X$ and latent representations $C_K, C_V$, we follow previous work (Ashkboos et al., 2024; Xiang & Zhang, 2024) and introduce two orthogonal matrices $S_1$ and $S_2$, and $S_1$ is also a Hadamard matrix with predefined binary elements. With these transformations, the quantized $Q, K, V$ computation for each head $h$ can be reformulated as:

$$Y_h = XA_hB_h \rightarrow Y_h = (XS_1^\top)(S_1A_hS_2^\top)(S_2B_h) \approx Q(XS_1^\top)Q(S_1A_hS_2^\top)Q(S_2B_h) \quad (10)$$

where $S_1^\top S_1 = S_2^\top S_2 = I$, $Q(\cdot)$ denotes the quantization operator, and we omit the QKV subscripts for simplicity of presentation. We further finetune the rotational matrices $S_2$ together with $A_h, B_h$ to minimize the change on the low-rank weights caused by quantization, with the objective as follows:

$$\min_{S_2,A_h,B_h} \left\| (F_h')^{1/2} \odot \left[ S_1W_h - Q(S_1A_hS_2^\top)Q(S_2B_h) \right] \right\|_2 \quad (11)$$

where $F_h' \approx \mathbb{E}_{x\sim\mathcal{D}}[(S_1g(x)) \odot (S_1g(x))]$. $F_h'$ is the Fisher information matrix associated with the transformed weight $S_1W_h$, computed element-wise as the root of the expected squared gradient $S_1g(x)$ over the calibration dataset $\mathcal{D}$. This acts as an importance weight, emphasizing parameters with higher sensitivity and guiding the QAT objective to more effectively preserve accuracy under quantization. During QAT, we jointly update $A_h, S_2$, and $B_h$, while $S_1$ is fixed as an exact Hadamard matrix of size $E \times E$, determined by the model embedding dimension $E$. This update design enables the factorized components to flexibly adapt to quantization noise while preserving the orthogonal transformation imposed by $S_1$, thereby maintaining the low-rank structure and improving the approximation accuracy and stability of low-precision training. Since this procedure is QAT performed locally, it incurs much lower time and memory overhead than end-to-end finetuning.

### 3.4 WSVD SYSTEM IMPLEMENTATION

A naive PyTorch implementation of WSVD results in excessive memory consumption during the reconstruction of $K$ and $V$, as illustrated in Figure 3 (a). Taking the key $K_h$ of head $h$ as an example, with $K_h = C_{Kh}B_{Kh}$ where $C_{Kh} \in \mathbb{R}^{L \times r}$ and $B_{Kh} \in \mathbb{R}^{r \times H}$, the GPU operation proceeds as follows. First, the latent representation $C_{Kh}$ is loaded from VRAM. Next, reconstruction $C_{Kh}B_{Kh}$ is performed, materializing the full $K_h \in \mathbb{R}^{L \times H}$ in VRAM. The reconstructed $K_h$ is then written back to VRAM and later reloaded for attention. Since $K_h$ and $V_h$ cannot fit into limited on-chip buffers, they must be stored along with the latent $C_{Kh}, C_{Vh}$, which largely increases I/O traffic and peak memory usage, in some cases exceeding that of the original model without low-rank decomposition.

To address this problem, we design a fused kernel in Triton (Tillet et al., 2019) that integrates low-rank reconstruction directly into the flash decoding pipeline, as shown in Figure 3 (b). At tile

granularity, the kernel streams a tile $t$ of $C_{Kh}$, denoted $C_{Kh,t} \in \mathbb{R}^{l \times r}$, from VRAM (step 1), where $l$ is the tile size along the sequence dimension $L$ that fits into on-chip memory. The up-projection weight $B_{Kh}$ is then loaded once into on-chip storage (step 2), and the temporary key tile $K_{h,t} = C_{Kh,t} B_{Kh}$ is formed in registers or shared memory (step 3). This process is executed within a single fused kernel that proceeds directly into the flash decoding pipeline: the temporary $K_{h,t}$ is immediately contracted with the query tile $q_h$ to compute $q_h K_{h,t}^\top$, update the online softmax statistics, and apply the normalized attention weights to the corresponding value tile $C_{Vh,t}$ (step 4).

In this design, both $C_{Kh}$ and $C_{Vh}$ are consumed in place, and all intermediate tensors remain on-chip without being written back to VRAM. The fused kernel integrates reconstruction, $qK^\top$ accumulation, softmax normalization, and the $V$ multiplication into a single workflow, eliminating redundant kernel launches and memory transfers. Memory usage now scales only with the tile size ($l \times r$ and $B_{Kh}$), which significantly reduces peak footprint and I/O traffic while preserving the efficiency of flash decoding. The design exposes parallelism at two levels: across tiles, where multiple tiles are processed concurrently within each head, and across heads, where different heads execute in parallel, fully utilizing GPU compute resources in accordance with flash decoding scheduling. Finally, the $V$-path up-projection $B_{Vh}$ is fused into the output projection, which avoids explicit reconstruction of $V_h$, following Palu (Chang et al., 2024). Collectively, these optimizations eliminate redundant memory operations while maintaining high parallelism, enabling WSVD to achieve practical inference acceleration without any loss of accuracy.

Beyond kernel fusion, WSVD applies per-head SVD to the Query, Key, and Value projections to reduce parameters and improve efficiency. Decomposing $W_K$ and $W_V$ decreases model size and accelerates both prefilling and decoding, while decomposing $W_Q$ further reduces parameters and speeds up prefilling. During prefilling, the input sequence is projected into low-rank $Q, K, V$ latents, with $K, V$ latents stored as cache.

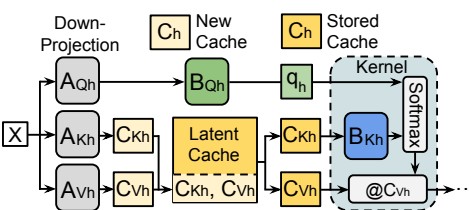

Figure 4: WSVD decoding pipeline. Each token is down-projected to low-rank latents, and $K$ and $V$ latents are appended to the cache, while $Q$ latent is up-projected and consumed together with cached $C_{Kh}, C_{Vh}$ in the fused kernel.

During the decoding stage, as shown in Figure 4, each new token is processed through per-head down-projections to generate low-rank latents for $Q$, $K$, and $V$. The latents of $K$ and $V$ are stored in the cache, while the latent of $Q$ is immediately up-projected to form $q$ for the current attention step. The kernel then loads the cached latents $C_{Kh}$ and $C_{Vh}$ together with the current $q_h$, performing highly parallelized computation that integrates low-rank reconstruction with flash decoding. This unified pipeline eliminates redundant materialization of full $K$ and $V$, preserves compact latent caches throughout decoding, and enables efficient attention computation with a reduced memory footprint.

## 4 EVALUATION

We conduct experiments on five representative vision–language models: LLaVA-v1.5 7B (Liu et al., 2023), LLaVA-v1.5 13B, LLaVA-Next 7B, LLaVA-Next 13B, and SmolVLM-Instruct (Marafioti et al., 2025). For local weighted fine-tuning and QAT, we use 256 samples randomly drawn from the ScienceQA training split (Lu et al., 2022), following the procedures described in Section 3.2 and Section 3.3. Evaluation is conducted on two widely used benchmarks, ScienceQA (Lu et al., 2022) and SEED-Bench-IMG (Li et al., 2024a), consistent with prior studies on VLMs such as LLaVA, using VLMEvalKit (Duan et al., 2024) tool. For comparison, WSVD is benchmarked against several baselines, including SVD-based approaches (ASVD (Yuan et al., 2023b), SVD-LLM (Wang et al., 2024d), QSVD (Wang et al., 2025d)) and quantization-based techniques (DuQuant (Lin et al., 2024), QVLM (Wang et al., 2024a)). For ASVD, SVD-LLM and QSVD, we follow their official implementations and apply SVD independently to the $Q, K, V$ matrices to ensure a fair comparison with WSVD, while leaving other linear layers unchanged. More results are shown in the Appendix.

To isolate the impact of SVD from quantization, we introduce **WSVD-noQ** (Section 3.2), which applies only the SVD techniques described in Sections 3.1 and 3.2. We compare it with ASVD, SVD-LLM, and QSVD-noQ (unquantized version of QSVD). We then apply QAT in Section 3.3 on

Table 1: Accuracy evaluation of different methods under FP16 (detailed results in Appendix A.3).

| Acc. | Method | ScienceQA-IMG ↑ | | | | | SEED-Bench ↑ | | | | | Avg. ↑ |
|---|---|---|---|---|---|---|---|---|---|---|---|---|
| | | $\rho_1:90\%$ | $\rho_1:80\%$ | $\rho_1:70\%$ | $\rho_1:60\%$ | $\rho_1:50\%$ | $\rho_1:90\%$ | $\rho_1:80\%$ | $\rho_1:70\%$ | $\rho_1:60\%$ | $\rho_1:50\%$ | |
| LLaVA-v1.5 7B | ASVD | 49.93% | 50.12% | 47.10% | 36.69% | 19.19% | 54.27% | 53.53% | 48.35% | 37.17% | 24.17% | 42.05% |
| | SVD-LLM | 65.44% | 63.71% | 61.92% | 57.41% | 55.53% | 57.89% | 57.50% | 55.33% | 54.64% | 55.31% | 58.47% |
| | QSVD-noQ | 67.72% | **68.22%** | 67.08% | 65.05% | 62.37% | 59.84% | 59.07% | 59.78% | 59.00% | 58.23% | 62.64% |
| | **WSVD-noQ** | **68.17%** | 67.72% | **67.28%** | **65.89%** | **65.49%** | **60.10%** | **60.17%** | **59.89%** | **60.18%** | **60.46%** | **63.54%** |
| | FP16 | Accuracy: 68.01% | | | | | Accuracy: 60.18% | | | | | 64.10% |
| LLaVA-Next 13B | ASVD | 71.24% | 70.60% | 71.44% | 71.38% | 69.81% | 70.88% | 70.26% | 70.01% | 69.69% | 69.01% | 70.43% |
| | SVD-LLM | 72.53% | 72.24% | 71.74% | 71.15% | 70.55% | 70.76% | 70.63% | 70.25% | 69.96% | 69.58% | 70.94% |
| | QSVD-noQ | 71.94% | 72.14% | 71.74% | 72.14% | 71.79% | 71.23% | 71.02% | 71.06% | 70.92% | 70.40% | 71.44% |
| | **WSVD-noQ** | **72.88%** | **72.98%** | **73.57%** | **73.48%** | **73.28%** | **71.29%** | **71.17%** | **71.25%** | **70.95%** | **70.81%** | **72.17%** |
| | FP16 | Accuracy: 73.23% | | | | | Accuracy: 71.30% | | | | | 72.27% |
| | | $\rho_1:90\%$ | | $\rho_1:80\%$ | | $\rho_1:70\%$ | $\rho_1:90\%$ | | $\rho_1:80\%$ | | $\rho_1:70\%$ | |
| SmolVLM 2B | ASVD | 29.30% | | 3.97% | | 0.20% | 17.85% | | 1.50% | | 0.95% | 8.96% |
| | SVD-LLM | 40.06% | | 17.20% | | 3.82% | 32.49% | | 15.89% | | 4.60% | 19.01% |
| | QSVD-noQ | **77.00%** | | 62.77% | | 42.59% | 64.80% | | 50.46% | | 36.24% | 55.64% |
| | **WSVD-noQ** | 76.30% | | **71.74%** | | **60.93%** | **65.78%** | | **63.29%** | | **54.45%** | **65.42%** |
| | FP16 | Accuracy: 84.53% | | | | | Accuracy: 68.47% | | | | | 76.53% |

top of WSVD-noQ, benchmarking against DuQuant, QVLM, and QSVD. We also evaluate QASVD, which applies QuaRot (Ashkboos et al., 2024) to the SVD-truncated VLMs produced by ASVD. For fine-tuning and QAT, we adopt lightweight local optimization to minimize overhead. $A_h$ and $B_h$ are updated with Adam (Kingma & Ba, 2014) (learning rate $1 \times 10^{-4}$ for fine-tuning and $1 \times 10^{-5}$ for QAT), while $S_2$ is updated during QAT using the Cayley optimizer (Wen & Yin, 2013). Local fine-tuning is performed for 100 steps and QAT for 50 steps, ensuring effective adaptation while keeping the additional latency negligible. All experiments are conducted on NVIDIA H100 GPUs.

## 4.1 ACCURACY EVALUATION ON WSVD-NOQ

We first evaluate the FP16 performance of WSVD-noQ under different rank budgets. To ensure fairness, we align the parameter ratio $\rho_1$ across all methods. For WSVD, $\rho_1$ is defined in Equation 3, while for other SVD-based baselines, $\rho_1$ is defined as the proportion of parameters relative to the original model after SVD is applied.

The evaluation results are summarized in Table 1 (details in Appendix A.3). Under the same parameter ratio $\rho_1$, WSVD-noQ surpasses ASVD, SVD-LLM, and QSVD-noQ in accuracy in most cases. On large-scale models such as LLaVA-v1.5 13B and LLaVA-Next 13B, WSVD-noQ incurs less than a 1% accuracy drop on ScienceQA-IMG and SEED-Bench compared to the FP16 baseline. Notably, for LLaVA-Next 13B, when $\rho_1 \leq 70\%$, WSVD-noQ even outperforms the FP16 model on ScienceQA-IMG. For example, at $\rho_1 = 70\%$, WSVD-noQ reaches 73.57% accuracy, exceeding the FP16 baseline by more than 0.3%. This suggests that low-rank approximation may implicitly mitigate hallucinations (Liu et al., 2024a), though further validation is required. Furthermore, WSVD-noQ delivers consistently higher average accuracy across datasets and parameter ratios. The advantage over other baselines becomes increasingly evident as $\rho_1$ decreases. For example, on SmolVLM, WSVD-noQ attains over 60% accuracy on ScienceQA-IMG, while baselines fail to yield usable results under the same parameter ratio settings.

## 4.2 ACCURACY EVALUATION OF WSVD

We present results under two weight–activation quantization configurations: W8A8 for WSVD with rank settings $\rho_1 = 50\%$ and $\rho_2 \approx 50\%$, and W8A4 for all other baselines. This design keeps cache size and parameter size comparable across methods, while WSVD's rank truncation further reduces its parameter budget, ensuring fairness in comparison.

For activation quantization, we adopt per-token symmetric quantization. For weight quantization, we employ round-to-nearest (RTN) with per-channel symmetric scaling and a learnable clipping ratio, where the clipping value is selected via linear search to minimize squared error, following QuaRot (Ashkboos et al., 2024). This quantization scheme is applied to the per-head Q/K/V weight matrices and all remaining attention and feed-forward modules, ensuring that the dominant matrix multiplications in each transformer block are executed in low precision. As shown in Table 2, WSVD consistently outperforms the baselines in most cases, despite using a smaller parameter budget and the same cache size. On average across models and datasets, WSVD incurs only a modest accuracy drop of just over 1% relative to the FP16 baseline, while reducing cache size to

Table 2: Accuracy evaluation of different methods under low-precision on LLaVA-v1.5 7B, LLaVA-v1.5 13B, LLaVA-Next 7B and LLaVA-Next 13B.

| Method | ScienceQA-IMG ↑ | | | | SEED-Bench ↑ | | | | Avg. ↑ |
|---|---|---|---|---|---|---|---|---|---|
| | v1.5 7B | v1.5 13B | Next 7B | Next 13B | v1.5 7B | v1.5 13B | Next 7B | Next 13B | |
| DuQuant | 57.36% | 67.22% | 66.34% | 70.20% | 54.11% | 61.43% | 63.64% | 66.15% | 63.31% |
| QVLM | 55.24% | 66.46% | 60.60% | 65.28% | 50.13% | 59.22% | 50.38% | 65.39% | 59.09% |
| QASVD | 41.92% | 65.34% | 49.37% | 64.85% | 41.26% | 59.30% | 49.63% | 66.54% | 54.78% |
| QSVD | **65.61%** | 70.12% | 66.10% | 70.43% | 58.49% | **62.95%** | 65.63% | 69.21% | 66.07% |
| **WSVD** | 64.25% | **72.14%** | **66.94%** | **73.08%** | **60.23%** | 62.01% | **67.49%** | **70.67%** | **67.10%** |
| FP16 | 68.10% | 71.83% | 69.60% | 73.23% | 60.18% | 62.54% | 69.02% | 71.30% | 68.23% |

Table 3: Results of weighted finetuning ablation under different $\rho_1$ settings.

| Acc. | Method | $\rho_1$=90% | $\rho_1$=70% | $\rho_1$=50% |
|---|---|---|---|---|
| v1.5 7B | FP16 | 68.01% | | |
| | WSVD-noFT | 67.82% | 66.82% | 65.09% |
| | WSVD-noQ | **68.17%** | **67.28%** | **65.49%** |

| Acc. | Method | $\rho_1$=90% | $\rho_1$=70% | $\rho_1$=50% |
|---|---|---|---|---|
| Next 7B | FP16 | 69.60% | | |
| | WSVD-noFT | 69.76% | 68.61% | 66.46% |
| | WSVD-noQ | **69.81%** | **69.36%** | **67.87%** |

25% of the FP16 model. At the same time, WSVD achieves more than 1% higher average accuracy than all baselines, demonstrating that the integration of per-head SVD and quantization with WSVD only lead to minimized accuracy loss.

## 4.3 ABLATION STUDY

**Effectiveness of Weighted Local Finetuning** We evaluate the impact of WSVD fine-tuning, as described in Section 3.2, on accuracy performance using ScienceQA-IMG for WSVD-noQ. The comparison is made against the WSVD-noQ baseline, which applies standard SVD without accounting for the relative importance of weight elements, while keeping all other settings identical. As shown in Table 3, WSVD-noQ consistently outperforms the no-finetuning variant (WSVD-noFT), demonstrating that incorporating relative weight importance during the SVD process leads to significantly improved performance over standard SVD.

**Effectiveness of QAT** We further examine the impact of local QAT on the low-rank model, as described in Section 3.3. Specifically, we compare WSVD against a baseline that uses the same quantization settings but does not fine-tune $S_2$, $A_h$, or $B_h$ mentioned in Section 3.3,

Table 4: Results of local QAT ablation.

| Method | ScienceQA-IMG ↑ | | | | Avg. ↑ |
|---|---|---|---|---|---|
| | v1.5 7B | v1.5 13B | Next 7B | Next 13B | |
| W/o QAT | 63.91% | 71.99% | 66.59% | 72.68% | 68.79% |
| WSVD | **64.25%** | **72.14%** | **66.94%** | **73.08%** | **69.10%** |

while keeping all other settings identical. As shown in Table 4, under W8A8, WSVD consistently surpasses the baseline across all models. These results demonstrate that local QAT effectively recovers the performance lost due to low-precision quantization.

## 4.4 SYSTEM EVALUATION ON VLM

We assess the system-level performance of WSVD-noQ, with a focus on decoding-stage acceleration. Specifically, we measure the layer-wise decoding latency of LLaVA-Next 7B across the attention and feed-forward modules using our fused kernel implementation described in Section 3.4 on RTX 4090 and 5090

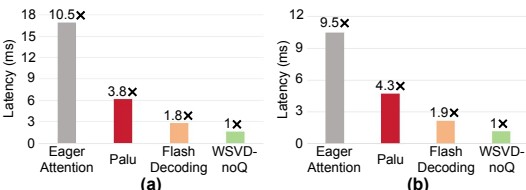

Figure 5: Latency evaluation and normalized latency on: (a) RTX 4090 and (b) RTX 5090.

GPUs. For comparison, we include Eager Attention without Flash Decoding, Palu (Chang et al., 2024), and Flash Decoding (Dao et al., 2023) as the baseline algorithms. For Flash Decoding, we adopt scaled dot-product attention (SDPA), while Palu is evaluated using its official repository. Both Eager Attention and Flash Decoding operate on the full KV cache, while Palu and WSVD-noQ restrict the latent size to $\rho_2 = 50\%$, corresponding to $\rho_1 \approx 51.5\%$ for WSVD. All measurements are conducted with a batch size of 16 and a sequence length of 8192. Since Palu supports only batch size 1, we use an equivalent sequence length of $16 \times 8192$ for fair comparison. In addition, we report latency results of full-matrix SVD and per-head SVD in Appendix A.7.

As shown in Figure 5, WSVD-noQ consistently outperforms all baselines on both GPUs in latency. Relative to Flash Decoding, it achieves over $1.8\times$ speedup, driven by reduced I/O overhead and negligible reconstruction cost enabled by our scheme. Compared with Palu, WSVD-noQ attains lower latency through two advantages: algorithmically, per-head SVD provides finer-grained compression than Palu's group-head SVD; system-wise, our fused kernel is fully integrated into the flash decoding pipeline. These results demonstrate that WSVD, together with our fused kernel design, offers an effective system-level solution that alleviates I/O bottlenecks and enables practical decoding acceleration in VLMs while maintaining accuracy performance as the original model.

Table 5: Latency (ms) on RTX 4090 (left) and RTX 5090 (right) for different sequence lengths.

| Seq Len | 1024 | 2048 | 4096 | 8192 | 16K | 32K | Seq Len | 1024 | 2048 | 4096 | 8192 | 16K | 32K |
|---|---|---|---|---|---|---|---|---|---|---|---|---|---|
| Flash Decoding | 0.92 | 1.21 | 1.77 | 2.92 | 5.14 | 9.64 | Flash Decoding | 0.65 | 0.86 | 1.28 | 2.14 | 3.81 | 7.18 |
| WSVD-noQ | 0.70 | 0.83 | 1.12 | 1.66 | 2.89 | 5.28 | WSVD-noQ | 0.58 | 0.66 | 0.83 | 1.15 | 1.79 | 3.06 |
| Speedup | 1.3× | 1.5× | 1.6× | 1.8× | 1.8× | 1.8× | Speedup | 1.1× | 1.3× | 1.5× | 1.9× | 2.1× | 2.3× |

Table 6: Latency (ms) on RTX 4090 (left) and RTX 5090 (right) for different batch sizes.

| Batch Size | 4 | 8 | 16 | 32 | 64 | Batch Size | 4 | 8 | 16 | 32 | 64 |
|---|---|---|---|---|---|---|---|---|---|---|---|
| Flash Decoding | 1.13 | 1.71 | 2.92 | 5.11 | 9.67 | Flash Decoding | 0.75 | 1.07 | 2.14 | 3.38 | 5.97 |
| WSVD-noQ | 0.82 | 1.11 | 1.66 | 2.91 | 5.33 | WSVD-noQ | 0.66 | 0.84 | 1.15 | 1.79 | 3.11 |
| Speedup | 1.4× | 1.5× | 1.8× | 1.8× | 1.8× | Speedup | 1.1× | 1.3× | 1.9× | 1.9× | 1.9× |

**Impact of Sequence Length and Batch Size** We further perform an ablation over various sequence lengths (Table 5) and batch sizes (Table 6) for LLaVA-Next 7B under the same setting, and report the layer-wise decoding latency of Flash Decoding and WSVD-noQ on RTX 4090 and RTX 5090 GPUs. With batch size 16, as the sequence length grows from 1K to 32K tokens, WSVD-noQ improves over Flash Decoding by about $1.3\times$ to $1.8\times$ on RTX 4090 and up to $2.35\times$ on RTX 5090. For a fixed 8192 sequence context, increasing batch size from 4 to 64 yields roughly $1.4\times$ to $1.9\times$ speedups on both GPUs. This trend reflects that longer contexts make KV-cache I/O increasingly dominant, so our WSVD-based compression and decoding kernel delivers larger relative gains.

**Impact of Rank Ratio** Using the same setting as Section 4.4, we vary the rank ratio $\rho_2 \in \{90\%, 70\%, 50\%\}$ for WSVD-noQ and measure the latency on RTX 4090 and RTX 5090 GPUs. Table 7 summarizes the impact of rank ratio on decoding latency. Smaller $\rho_2$ values (i.e., lower ranks) consistently yield lower latency, demonstrating that WSVD's fused kernel can effectively translate rank reduction into tangible decoding speedups over the Flash Decoding baseline.

Table 7: Latency (ms) under different $\rho_2$.

| GPU | Flash Dec. | $\rho_2$:90% | $\rho_2$:70% | $\rho_2$:50% |
|---|---|---|---|---|
| 4090 | 2.92 | 2.83 | 2.53 | 1.66 |
| 5090 | 2.14 | 1.87 | 1.75 | 1.15 |

Table 8: Latency (ms) on RTX 3060 ($\rho_2 : 50\%$).

| Seq Len | 1024 | 2048 | 4096 | 8192 | 16K |
|---|---|---|---|---|---|
| Flash Decoding | 3.37 | 4.88 | 7.81 | 13.27 | 24.59 |
| WSVD-noQ | 2.18 | 2.68 | 3.62 | 5.54 | 9.49 |
| Speedup | 1.5× | 1.8× | 2.2× | 2.4× | 2.6× |

**Speedup on Low-end GPU** To evaluate our method on more modest hardware, we benchmark the latency of LLaVA-Next 7B with WSVD-noQ on an RTX 3060 (Table 8) under the same setting, and compare it with the Flash Decoding baseline. On RTX 3060, WSVD-noQ reduces latency from 3.37 ms to 2.18 ms at 1K tokens ($1.55\times$) and from 24.59 ms to 9.49 ms at 16K tokens ($2.59\times$). These speedups are larger than on 4090/5090-class GPUs because the lower memory bandwidth of RTX 3060 makes KV-cache I/O more dominant. By shrinking the KV cache and using a fused decoding kernel, WSVD reduces memory traffic and achieves larger latency gains on low-end devices.

## 5 CONCLUSION

In this work, we present WSVD, a weighted low-rank approximation framework that integrates per-head SVD, weighted fine-tuning, and quantization-aware training to compress and accelerate vision–language models. By aligning algorithmic design with system-level optimization through our fused kernel, WSVD achieves over $1.8\times$ decoding speedup while preserving accuracy under aggressive compression.

ETHICS STATEMENT

This work focuses on model compression and acceleration techniques for vision–language models. All datasets used in this study (ScienceQA-IMG and SEED-Bench) are publicly available and widely adopted in the community. Our research does not involve human subjects, private or sensitive data, or personally identifiable information. The proposed method aims to improve the efficiency of large models, which may contribute to reducing the computational and environmental costs of deployment. We are not aware of any direct ethical concerns specific to this work.

REPRODUCIBILITY STATEMENT

We take reproducibility seriously and provide the following details:

- **Code and models:** We will release the full implementation of WSVD, including training and inference code, as well as evaluation scripts, upon publication.

- **Datasets:** All datasets used in this work are publicly available. In particular, we evaluate on ScienceQA-IMG and SEED-Bench, both of which can be accessed without restriction. We will also provide preprocessing scripts to reproduce the exact input formats used in our experiments.

- **Experimental setup:** Hyperparameters, optimization settings, and evaluation protocols are described in detail in Section 4.

- **Randomness:** All experiments are run with fixed random seeds in the scripts, to ensure consistent results.

- **Compute resources:** Our experiments are conducted on NVIDIA H100, RTX 4090 and RTX 5090 GPUs as described in Section 4.

- **Limitations:** Some large-scale experiments (e.g., on 13B-parameter models) require access to high-end GPUs, which may limit reproducibility for groups without such resources.

We believe that with the released code, scripts, and dataset accessibility, other researchers will be able to reproduce our results and build upon our method.

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

# A APPENDIX

## A.1 THE USE OF LLMS

Large language models (LLMs), such as ChatGPT, were used exclusively for language polishing and minor stylistic editing of the manuscript. All technical ideas, analyses, and experimental results were conceived, implemented, and verified by the authors. The authors carefully reviewed and validated all text to ensure accuracy.

## A.2 WSVD ALGORITHM

---

**Algorithm 1** Weighted SVD Fine-tuning (WSVD) pseudo code

---

**Require:** Calibration dataset $X$, model parameters $\{W\}$, rank $r$
**Ensure:** Fine-tuned low-rank factors $\{A, B\}$
 1: **for** each sample $x_i \in X$ **do**
 2:     Compute forward pass and loss $\mathcal{L}(x_i)$
 3:     Backpropagate to obtain gradients $\{\nabla_W \mathcal{L}\}$
 4:     Accumulate importance weights $F \leftarrow \sum_X (\nabla_W \mathcal{L})^2$
 5: **end for**
 6: **for** each weight matrix $W \in \{W_Q, W_K, W_V, \dots\}$ **do**
 7:     Compute SVD: $W_h \approx A_h B_h$, with $A_h \in \mathbb{R}^{m \times r}, B_h \in \mathbb{R}^{r \times n}$
 8:     Define weighted loss:

$$\mathcal{L}_{\text{WSVD}}(A_h, B_h) = \left\| F_h^{1/2} \odot (W_h - A_h B_h) \right\|_F^2$$

    where $F_h, W_h, A_h$ and $B_h$ is each head's importance weight, weight, decomposed matrices.
 9:     Locally fine-tune $A_h, B_h$ using $\mathcal{L}_{\text{WSVD}}$
10: **end for**
11: **return** Fine-tuned low-rank factors $\{A_h, B_h\}$ for all matrices

---

## A.3 DETAILED RESULTS OF WSVD-NOQ

We inlcude the detailed results of WSVD-noQ and other baselines in Table 9.

Table 9: Accuracy evaluation of different methods under FP16.

| Acc. | Method | ScienceQA-IMG ↑ | | | | | SEED-Bench ↑ | | | | | Avg. ↑ |
|---|---|---|---|---|---|---|---|---|---|---|---|---|
| | | $\rho_1:90\%$ | $\rho_1:80\%$ | $\rho_1:70\%$ | $\rho_1:60\%$ | $\rho_1:50\%$ | $\rho_1:90\%$ | $\rho_1:80\%$ | $\rho_1:70\%$ | $\rho_1:60\%$ | $\rho_1:50\%$ | |
| LLaVA-v1.5 7B | ASVD | 49.93% | 50.12% | 47.10% | 36.69% | 19.19% | 54.27% | 53.53% | 48.35% | 37.17% | 24.17% | 42.05% |
| | SVD-LLM | 65.44% | 63.71% | 61.92% | 57.41% | 55.53% | 57.89% | 57.50% | 55.33% | 54.64% | 55.31% | 58.47% |
| | QSVD-noQ | 67.72% | **68.22%** | 67.08% | 65.05% | 62.37% | 59.84% | 59.07% | 59.78% | 59.00% | 58.23% | 62.64% |
| | **WSVD-noQ** | **68.17%** | 67.72% | **67.28%** | **65.89%** | **65.49%** | **60.10%** | **60.17%** | **59.89%** | **60.18%** | **60.46%** | **63.54%** |
| | FP16 | Accuracy: 68.01% | | | | | Accuracy: 60.18% | | | | | 64.10% |
| LLaVA-v1.5 13B | ASVD | 71.39% | 71.59% | 70.00% | 70.25% | 69.51% | 61.92% | 61.91% | 61.54% | 61.51% | 60.71% | 66.03% |
| | SVD-LLM | 71.05% | 70.85% | 70.30% | 70.35% | 70.30% | 62.28% | 62.34% | 62.25% | 62.08% | **63.01%** | 66.48% |
| | QSVD-noQ | 71.89% | **71.99%** | 71.49% | 71.54% | 71.39% | **62.61%** | 62.64% | **62.82%** | **62.63%** | 62.52% | 67.15% |
| | **WSVD-noQ** | **71.99%** | 71.84% | **72.53%** | **71.59%** | **71.44%** | 62.52% | **62.68%** | 62.38% | 62.37% | 62.37% | **67.17%** |
| | FP16 | Accuracy: 71.83% | | | | | Accuracy: 62.53% | | | | | 67.18% |
| LLaVA-Next 7B | ASVD | 64.20% | 63.36% | 62.07% | 60.19% | 55.28% | 67.38% | 66.96% | 66.24% | 65.13% | 61.52% | 63.23% |
| | SVD-LLM | 68.27% | 67.92% | 66.58% | 66.39% | 65.54% | 68.50% | 68.31% | 67.65% | 67.45% | 66.28% | 67.29% |
| | QSVD-noQ | **70.10%** | 69.16% | 69.01% | **68.27%** | 66.19% | 68.86% | 68.95% | 68.44% | 67.98% | 67.27% | 68.42% |
| | **WSVD-noQ** | 69.81% | **69.56%** | **69.36%** | 68.22% | **67.87%** | **69.18%** | **69.27%** | **69.15%** | **69.16%** | **68.59%** | **69.02%** |
| | FP16 | Accuracy: 69.60% | | | | | Accuracy: 69.02% | | | | | 69.31% |
| LLaVA-Next 13B | ASVD | 71.24% | 70.60% | 71.44% | 71.38% | 69.81% | 70.88% | 70.26% | 70.01% | 69.69% | 69.01% | 70.43% |
| | SVD-LLM | 72.53% | 72.24% | 71.74% | 71.15% | 70.55% | 70.76% | 70.63% | 70.25% | 69.96% | 69.58% | 70.94% |
| | QSVD-noQ | 71.94% | 72.14% | 71.74% | 72.14% | 71.79% | 71.23% | 71.02% | 71.06% | 70.92% | 70.40% | 71.44% |
| | **WSVD-noQ** | **72.88%** | **72.98%** | **73.57%** | **73.48%** | **73.28%** | **71.29%** | **71.17%** | **71.25%** | **70.95%** | **70.81%** | **72.17%** |
| | FP16 | Accuracy: 73.23% | | | | | Accuracy: 71.30% | | | | | 72.27% |
| SmolVLM 2B | | $\rho_1:90\%$ | $\rho_1:80\%$ | $\rho_1:70\%$ | | | $\rho_1:90\%$ | $\rho_1:80\%$ | $\rho_1:70\%$ | | | |
| | ASVD | 29.30% | 3.97% | 0.20% | | | 17.85% | 1.50% | 0.95% | | | 8.96% |
| | SVD-LLM | 40.06% | 17.20% | 3.82% | | | 32.49% | 15.89% | 4.60% | | | 19.01% |
| | QSVD-noQ | **77.00%** | 62.77% | 42.59% | | | 64.80% | 50.46% | 36.24% | | | 55.64% |
| | **WSVD-noQ** | 76.30% | **71.74%** | **60.93%** | | | **65.78%** | **63.29%** | **54.45%** | | | **65.42%** |
| | FP16 | Accuracy: 84.53% | | | | | Accuracy: 68.47% | | | | | 76.53% |

## A.4 COMPARISON WITH DOBISVD

For completeness, we additionally evaluate DobiSVD on LLaVA-v1.5 7B using the ScienceQA-IMG benchmark. Following the official implementation, DobiSVD is applied to the $Q, K, V$ matrices while leaving other linear layers unchanged. Calibration is performed using the same set of samples and initialized with the same random seed as in the main experiments to ensure fairness. As shown in Table 10, WSVD-noQ achieves higher accuracy than DobiSVD under the same compression ratio.

Table 10: Accuracy evaluation of different methods under FP16.

| Acc. | Method | ScienceQA-IMG ↑ | | | | |
| | | $\rho_1 : 90\%$ | $\rho_1 : 80\%$ | $\rho_1 : 70\%$ | $\rho_1 : 60\%$ | $\rho_1 : 50\%$ |
|---|---|---|---|---|---|---|
| LLaVA-v1.5 7B | DobiSVD | 67.19% | 60.94% | 59.38% | 56.64% | 54.69% |
| | ASVD | 49.93% | 50.12% | 47.10% | 36.69% | 19.19% |
| | SVD-LLM | 65.44% | 63.71% | 61.92% | 57.41% | 55.53% |
| | QSVD-noQ | 67.72% | **68.22%** | 67.08% | 65.05% | 62.37% |
| | **WSVD-noQ** | **68.17%** | 67.72% | **67.28%** | **65.89%** | **65.49%** |
| | FP16 | Accuracy: 68.01% | | | | |

## A.5 SUPPLEMENTARY RESULTS ON MORE VLMS

We additionally apply WSVD to Qwen-VL 7B (Bai et al., 2023) and Molmo-7B-O (Deitke et al., 2025) to further examine the generality of WSVD beyond the LLaVA family and SmolVLM. We follow exactly the same experimental setting as Section 4 and evaluate the FP16 and SVD compressed models on ScienceQA-IMG and SEED-Bench. As shown in Table 11, WSVD-noQ consistently outperforms all SVD-based baselines (ASVD and SVDLLM) across all singular-value ratios, and it also matches or slightly improves over the FP16 model. For example, on ScienceQA, WSVD-noQ improves over SVDLLM and ASVD by up to 3–5%, and on SEED-Bench it yields the best/comparable accuracy among all compressed variants at every ratio. These results indicate that WSVD transfers well to VLMs with different vision–language fusion designs, supporting the general applicability of our method.

Table 11: Accuracy evaluation of different methods under FP16 on Qwen-VL 7B and Molmo-7B-O.

| Acc. | Method | ScienceQA-IMG ↑ | | | | SEED-Bench ↑ | | | | Avg. ↑ |
| | | $\rho_1 : 90\%$ | $\rho_1 : 80\%$ | $\rho_1 : 70\%$ | $\rho_1 : 60\%$ | $\rho_1 : 90\%$ | $\rho_1 : 80\%$ | $\rho_1 : 70\%$ | $\rho_1 : 60\%$ | |
|---|---|---|---|---|---|---|---|---|---|---|
| Qwen-VL 7B | ASVD | 63.86% | 63.11% | 60.54% | 59.49% | 61.67% | 61.00% | 59.94% | 58.69% | 61.04% |
| | SVD-LLM | 65.29% | 65.29% | 64.55% | 65.29% | 62.99% | 62.80% | 62.69% | 62.51% | 63.93% |
| | QSVD-noQ | 66.78% | 66.24% | 66.68% | 65.20% | 63.25% | 63.00% | 62.06% | 61.26% | 64.31% |
| | **WSVD-noQ** | **68.77%** | **68.12%** | **67.23%** | **65.49%** | **63.32%** | **63.53%** | **63.60%** | **63.68%** | **65.47%** |
| | FP16 | Accuracy: 68.32% | | | | Accuracy: 63.52% | | | | 65.92% |
| Molmo 7B-O | ASVD | 94.99% | 95.19% | 94.10% | 93.06% | 74.42% | 74.21% | 73.73% | 73.62% | 84.17% |
| | SVD-LLM | 95.09% | 95.09% | 94.99% | **95.09%** | **74.68%** | 74.46% | 74.23% | 74.11% | 84.72% |
| | QSVD-noQ | 95.54% | 94.99% | 94.59% | 93.85% | 74.47% | 74.44% | 74.41% | 74.37% | 84.58% |
| | **WSVD-noQ** | **95.59%** | **95.49%** | **95.34%** | **95.09%** | 74.61% | **74.52%** | **74.48%** | **74.38%** | **84.94%** |
| | FP16 | Accuracy: 95.78% | | | | Accuracy: 74.74% | | | | 85.26% |

## A.6 SUPPLEMENTARY RESULTS ON MORE DATASETS

To further assess generalization, we additionally evaluate LLaVA-Next 13B with WSVD-noQ on two additional benchmarks: HRBench-4K (4K high-resolution images) (Wang et al., 2025b), and OCRBench (text-centric images) (Liu et al., 2024c). Following Section 4, we reuse the same 256-sample calibration set drawn from the ScienceQA training set and keep all other settings identical, while sweeping the parameter ratios $\rho_1$. As summarized in Table 12, WSVD-noQ consistently matches or outperforms all baselines across nearly all ratios on these datasets, despite being calibrated only once on the ScienceQA training set. These results indicate that WSVD generalizes well across tasks and datasets. Moreover, WSVD's decoding speedup is independent of the evaluation dataset: once the model is calibrated and compressed, runtime is determined solely by the resulting model size and context length, so a fixed compressed model yields essentially the same speedup across benchmarks.

Table 12: Accuracy evaluation of different methods under FP16 on OCRBench and HRBench-4K.

| Acc. | Method | OCRBench ↑ | | | | | HRBench-4K ↑ | | | | | Avg. ↑ |
|---|---|---|---|---|---|---|---|---|---|---|---|---|
| | | $\rho_1:90\%$ | $\rho_1:80\%$ | $\rho_1:70\%$ | $\rho_1:60\%$ | $\rho_1:50\%$ | $\rho_1:90\%$ | $\rho_1:80\%$ | $\rho_1:70\%$ | $\rho_1:60\%$ | $\rho_1:50\%$ | |
| LLaVA-Next 13B | ASVD | 53.10% | 52.50% | 51.30% | 50.50% | 47.70% | 44.00% | 44.13% | 43.88% | 43.00% | 42.00% | 47.21% |
| | SVD-LLM | 52.10% | 51.90% | 51.00% | 49.90% | 48.20% | 43.00% | 44.25% | 42.62% | 43.75% | 43.25% | 47.00% |
| | QSVD-noQ | 52.80% | 52.40% | 52.40% | 51.40% | **48.90%** | 44.25% | **44.88%** | 43.88% | 43.37% | 42.88% | 47.72% |
| | **WSVD-noQ** | **53.30%** | **53.30%** | **53.50%** | **52.20%** | 48.70% | **46.13%** | **44.88%** | **44.50%** | **44.88%** | **44.50%** | **48.59%** |
| | FP16 | Accuracy: 53.30% | | | | | Accuracy: 45.63% | | | | | 49.47% |

## A.7 LATENCY COMPARISON OF FULL AND PER-HEAD SVD

We further compare the decoding latency of applying SVD to the full QKV matrices versus adopting WSVD's fine-grained per-head SVD. To enable this comparison, we minimally modify our kernel to support reconstruction with larger matrix sizes under the full SVD setting (as discussed in Section 3.2), while still fusing the reconstruction with flash decoding. This variant is denoted as "W/o per-head." Both approaches are evaluated under the same $\rho_2$, ensuring equal overall cache size, with batch size, sequence length and other settings kept identical to the setup described above.

Table 13: Decoding latency on RTX 4090.

| $\rho_2$ | W/o per-head | WSVD-noQ | Speedup |
|---|---|---|---|
| 90% | 51.31 | 2.83 | 18.1× |
| 80% | 46.03 | 2.60 | 17.7× |
| 70% | 39.46 | 2.53 | 15.6× |
| 60% | 33.54 | 2.25 | 14.9× |
| 50% | 28.37 | 1.66 | 17.1× |

Table 14: Decoding latency on RTX 5090.

| $\rho_2$ | W/o per-head | WSVD-noQ | Speedup |
|---|---|---|---|
| 90% | 40.40 | 1.87 | 21.6× |
| 80% | 35.99 | 1.77 | 20.3× |
| 70% | 31.14 | 1.75 | 17.8× |
| 60% | 26.85 | 1.56 | 17.2× |
| 50% | 21.44 | 1.15 | 18.6× |

As shown in Tables 13 and 14, WSVD-noQ consistently achieves more than an order-of-magnitude speedup over the full-matrix SVD variant ("W/o per-head") across all compression ratios $\rho_2$. On RTX 4090, the speedup ranges from $14.9\times$ to $18.1\times$, while on RTX 5090 it further increases to $17.2\times$–$21.6\times$. These results confirm that per-head SVD substantially reduces reconstruction overhead and I/O traffic, enabling efficient decoding.

## A.8 TRAINING COST OF WSVD

WSVD first applies SVDLLM's whitening method (Wang et al., 2024d) to per-head weight matrices before performing SVD, then uses QSVD's importance-score-based rank allocation (Wang et al., 2025d) to truncate the model, and subsequently performs lightweight, Fisher-information-based local fine-tuning and local quantization-aware training on the truncated low-rank weights to better preserve the most sensitive weight elements and to mitigate the degradation of per-head SVD and low-precision inference.

Table 15: Calibration time breakdown for QSVD and WSVD.

| QSVD | | WSVD | |
|---|---|---|---|
| Step | Time | Step | Time |
| Input calibration | 1 min | Input calibration | 6 min |
| SVD on all layers | 2 min 15 s | SVD on all layers | 1 min |
| Gradient collection & rank allocation | 10 min 12 s | Gradient collection & rank allocation | 13 min 40 s |
| SVD results fusion | 30 s | Local FT | 9 min |
| $\beta$ tuning & quantization | 82 min | Local QAT & quantization | 8 min |
| Total calibration time | 96 min | Total calibration time | 38 min |

We quantify the computational overhead of WSVD's local fine-tuning and QAT and compare it with QSVD on LLaVA-1.5 13B. We extract QSVD's reported training time from its OpenReview page and, for fairness, benchmark WSVD on the same GPU type (A100). As shown in Table 15, QSVD requires **96 minutes ≈ 1.6 A100 GPU-hours**, whereas WSVD takes only **38 minutes ≈ 0.63 A100 GPU-hours**, about **2.5× less tuning time**. The peak GPU memory usage of WSVD's

local FT and QAT stages on LLaVA-1.5 13B is only **15 GB**, since we only perform local updates on low-rank weights rather than end-to-end fine-tuning and do not store full intermediate activations, so the whole procedure fits comfortably on a single A100-80GB. For additional context, the official LLaVA-1.5 report (Liu et al., 2024b) states that training the 13B model requires at least **204 A100 GPU-hours**. Thus, WSVD's tuning cost is only a small fraction of the original training cost, while still delivering practical decoding speedups, indicating that the efficiency gains comfortably justify the modest local fine-tuning and QAT overhead.

