# OpenReview forum: "WSVD: Weighted Low-Rank Approximation for Fast and Efficient Execution of Low-Precision Vision-Language Models"
_ICLR.cc/2026/Conference — ICLR 2026 Poster_

### Official Review · Reviewer_YkZ1 · 2025-10-29

**Soundness:** 3
**Presentation:** 3
**Contribution:** 2
**Rating:** 4
**Confidence:** 2

**Summary:**

This paper proposes a method that accelerates vision-language model inference by combining Weighted Singular Value Decomposition (WSVD) with low-precision quantization. Unlike standard SVD-based compression, WSVD assigns importance weights to parameters to better preserve critical information during low-rank approximation. The approach further integrates quantization, leading to significant latency reduction, reportedly achieving faster decoding with minimal accuracy loss. Overall, WSVD aims to offer a practical and effective way to deploy large VLMs more efficiently without sacrificing performance.

**Strengths:**

- The paper presents a practical and well-motivated approach to enhancing the efficiency of vision-language models. In particular, it provides a comprehensive exploration of memory reduction strategies from multiple perspectives, including low-rank decomposition and quantization, and proposes a corresponding system implementation to support these ideas.
- The paper is clearly written and well organized, making the contributions easy to follow.

**Weaknesses:**

I'm not very familiar with this specific research area, such as quantization and system implementation. Therefore, it is difficult for me to precisely assess the novelty of the proposed method relative to existing work.

- From my perspective, the contribution over prior work appears incremental. Please refer to the question below for clarification.
- The experimental validation is limited to only two datasets, and broader comparisons across more diverse datasets are missing.
- The related work section omits several important studies on weighted matrix factorization, which weakens the paper’s novelty claim.

**Questions:**

- The computational cost of reconstruction for each head is $LrH$. Wouldn’t it be equivalent to simply reducing $R$ to $r$ in the original $LRH$ formulation? How is this approach fundamentally different?
- Why is the square root used in the Fisher matrix in Equation (8)?
- There exist several established methods for measuring parameter importance [1, 2]. Why were these approaches not explored or compared? Is the proposed method guaranteed to be optimal?
- The paper states that FWSVD utilizes Fisher information. How exactly do FWSVD and WSVD differ in the way they incorporate Fisher information?

References\
[1] Ancona, M., Ceolini, E., Öztireli, C., & Gross, M. (2017). Towards better understanding of gradient-based attribution methods for deep neural networks. arXiv preprint arXiv:1711.06104.\
[2] Bach, S., Binder, A., Montavon, G., Klauschen, F., Müller, K. R., & Samek, W. (2015). On pixel-wise explanations for non-linear classifier decisions by layer-wise relevance propagation. PloS one, 10(7), e0130140.

---

> ### Author Response · Authors · 2025-11-22
>
> We are grateful for your constructive comments. The following section summarizes your main concerns and questions, along with our detailed replies.
>
> 1. **The computational cost of reconstruction for each head is $LrH$. Wouldn’t it be equivalent to simply reducing $R$ to $r$  in the original $LRH$ formulation? How is this approach **fundamentally different**?**
>
>     This reconstruction pattern is **fundamentally different** from prior SVD-based compression, as detailed in Section 3.1. Although the reconstruction cost per head in WSVD, $L r H$, looks similar to the original $L R H$ expression with $R$ replaced by $r$, $R$ and $r$ are fundamentally different ranks. Prior SVD methods such as SVDLLM and QSVD apply SVD to the full (or concatenated) projection $W_K \in \mathbb{R}^{E \times E}$, so $R$ is chosen at the **model-dimension** scale $E$ (thousands). In contrast, WSVD performs **per-head** SVD on $W_{K h} \in \mathbb{R}^{E \times H}$, so $r$ is chosen at the much smaller **head-dimension** scale $H$ (10x).
>
>     To compare reconstruction computational cost fairly, we align the relative rank used in each factorization: full-matrix SVD retains an $R/E$ fraction of the singular values of $W_K$, while WSVD retains an $r/H$ fraction per head. Since $E = N_{\text{heads}} \cdot H$, matching the same relative rank implies
>
>      $\frac{R}{E} = \frac{r}{H}$
>
>     $\Rightarrow R = \frac{E}{H} r = N_{\text{heads}} \cdot r$.
>
>     Under this alignment, the reconstruction cost ratio becomes
>
>     $\frac{L r H}{L R H} = \frac{r}{R} = \frac{1}{N_{\text{heads}}}$,
>
>     so WSVD reduces reconstruction compute by roughly a factor of $N_{\text{heads}}$ (e.g., $\sim 32\times$ for LLaVA-Next 7B). Simply “reducing $R$ to $r$” in the original $L R H$ formulation would correspond to an extremely aggressive global rank (i.e., $R/E \ll r/H$), leading to severe accuracy loss and therefore is not comparable to our setting.
>
>     Moreover, the computation pattern is different. Full-matrix SVD stores a large shared latent $C_K \in \mathbb{R}^{L \times R}$ that every head must repeatedly load, resulting in high KV-cache I/O. WSVD instead stores small per-head latents $C_{K h} \in \mathbb{R}^{L \times r}$ and fuses their reconstruction with attention in our fused kernel (Section 3.4), which further reduces memory traffic. In Appendix A.5 we show that, even at the same $\rho_2$ and with a fused kernel, full-matrix SVD (**w/o per-head**) is $15$-$22\times$ slower than WSVD. Hence WSVD is not just a smaller rank in the same formulation, but a different SVD granularity and system design that yields an order-of-magnitude drop in reconstruction cost.
>
> 2. **Why is the square root used in the Fisher matrix in Equation (8)?**
>
>     The square root in Eq. (8) comes directly from the second–order Fisher-based approximation in Eqs. (6)–(7). From a second-order Taylor expansion, the expected loss change induced by a perturbation $\Delta W_K$ is
>
>     $\Delta L \approx \tfrac{1}{2} \sum_{i,j} F_{K(i,j)} (\Delta W_{K(i,j)})^2$,
>
>     where $F_{K(i,j)}$ is the Fisher information score for the element $(i,j)$ of parameter $W_K$ as defined in Eq. (7). To make the SVD objective consistent with this approximation, we want the weighted reconstruction error to be
>
>     $\sum_{i,j} F_{K(i,j)} \bigl(W_{K(i,j)} - (A_K B_K)_{(i,j)}\bigr)^2$
>
>     Writing this in matrix form yields
>
>     $\sum_{i,j} F_{K(i,j)} \bigl(W_{K(i,j)} - (A_K B_K)_{(i,j)}\bigr)^2$
>
>     $= \bigl\|\bigl\|  F_K^{1/2} \odot (W_K - A_K B_K) \bigr\|\bigr\|_F^2$,
>
>     since the Frobenius norm already squares its argument element-wise. Thus the Fisher weights $F_{K(i,j)}$ appear inside the squared objective, and placing $F_K^{1/2}$ inside the norm is simply a notational convenience to "absorb" $F_K$ into the square, rather than an extra modification. In other words, Eq. (8) is exactly the Fisher-weighted squared error $\sum_{i,j} F_{K(i,j)} \bigl(W_{K(i,j)} - (A_K B_K)_{(i,j)}\bigr)^2$, just rewritten using $F_K^{1/2}$ so that the objective remains a standard Frobenius norm.

---

> > ### Author Response · Authors · 2025-11-22
> >
> > 3. **There exist several established methods for measuring parameter importance [1, 2]. Why were these approaches not explored or compared? Is the proposed method guaranteed to be optimal?**
> >
> >     The cited methods [1,2] focus on *input attribution* and compute relevance scores with respect to the **input or intermediate activations**, rather than the model parameters. In contrast, our method directly analyzes the sensitivity of the model output to its **parameters**, deriving a principled influence score that is suitable for structured compression and low-rank reconstruction. Because these prior works do not provide parameter-level importance measures, we replace activation to weight for a simple comparison. Below we provide the integrated grad and input*grad method comparison mentioned in the reference paper, overall, our fisher information based importance delivers the best results.
> >
> >     | P ratio | grad | fisher | integrated grad | input*grad |
> >     | --- | --- | --- | --- | --- |
> >     | 0.9 | 66.14 | **68.17** | 67.97 | 67.54 |
> >     | 0.8 | 66.78 | **67.72** | 54.09 | 53.39 |
> >     | 0.7 | 64.80 | **67.28** | 45.56 | 45.12 |
> >     | 0.6 | 63.96 | **65.89** | 39.76 | 39.06 |
> >     | 0.5 | 63.36 | **65.49** | 31.28 | 31.63 |
> >
> >     Our method is not guaranteed to be globally optimal, as optimal parameter ranking under rank constraints is generally intractable, but it provides a theoretically grounded second-order approximation that performs strongly in practice.
> >
> >     [1] Ancona, M., Ceolini, E., Öztireli, C., & Gross, M. (2017). Towards better understanding of gradient-based attribution methods for deep neural networks. arXiv preprint arXiv:1711.06104.
> >
> >     [2] Bach, S., Binder, A., Montavon, G., Klauschen, F., Müller, K. R., & Samek, W. (2015). On pixel-wise explanations for non-linear classifier decisions by layer-wise relevance propagation. PloS one, 10(7), e0130140.
> > 4. **The paper states that FWSVD utilizes Fisher information. How exactly do FWSVD and WSVD differ in the way they incorporate Fisher information?**
> >
> >     We note that ASVD Section A.6 [1] and SVDLLM Section 4.1 [2] already compare against FWSVD in their original papers and consistently outperform it. Following this practice, we do not treat FWSVD as a primary baseline and instead use ASVD and SVDLLM as stronger SVD-based baselines. As shown in Table 1, WSVD further outperforms both ASVD and SVDLLM.
> >
> >     Both FWSVD and WSVD leverage Fisher information, but at different granularities and stages. FWSVD [3] starts from element-wise Fisher scores $\hat{I}\_\{W_\{ij\}\}$, aggregates them into a single row-wise importance $\hat{I}\_{W_i} = \sum_j \hat{I}\_{W\_{ij}}$, and optimizes $\min\_{A,B} \|\|\hat{I}W - \hat{I}AB\|\|\_F^2$ with $\hat{I} = \mathrm{diag}(\sqrt{\hat{I}\_{W\_1}},\dots,\sqrt{\hat{I}\_{W\_N}})$. This reduces to applying standard SVD to the pre-scaled matrix $\hat{I}W$, so all weights in a row share one Fisher weight and Fisher is used only once as a pre-scaling step.
> >
> >     In contrast, WSVD uses Fisher at a finer, element-wise level and in two stages. For each head $h$, we compute $F_h \in \mathbb{R}^{E \times H}$ and solve $\min_{A_{Kh},B_{Kh}} \|\|F_h^{1/2} \odot (W_{Kh} - A_{Kh}B_{Kh})\|\|_F^2$ (Equation (9)), without collapsing Fisher scores across rows or columns. During QAT, we use another element-wise Fisher matrix $F'_h$ in the transformed space (Eq. (11)) to weight quantization error and guide the updates of $A_h$, $S_2$, and $B_h$. In summary, FWSVD applies Fisher in a coarse, row-wise manner before a closed-form SVD, whereas WSVD applies Fisher element-wise and per head throughout both local fine-tuning and QAT, enabling a more fine-grained and effective use of Fisher information.
> >
> >     [1] Yuan, Zhihang, et al. "Asvd: Activation-aware singular value decomposition for compressing large language models." arXiv preprint arXiv:2312.05821 (2023).
> >
> >     [2] Wang, Xin, et al. "Svd-llm: Truncation-aware singular value decomposition for large language model compression." arXiv preprint arXiv:2403.07378 (2024).
> >
> >     [3] Hsu, Yen-Chang, et al. "Language model compression with weighted low-rank factorization." arXiv preprint arXiv:2207.00112 (2022).
> >
> > 5. **The experimental validation is limited to only two datasets.**
> >
> >     To further assess generalization, the revised paper evaluates WSVD-noQ on two additional VLMs (**Appendix A.5**) and two additional benchmarks (**Appendix A.6**). Across these models and datasets, WSVD-noQ consistently matches or outperforms all baselines under comparable rank budgets, indicating that WSVD generalizes well across architectures and tasks. Note that WSVD’s decoding speedup is independent of the evaluation dataset: once the model is calibrated and compressed, runtime is determined mainly by the resulting model size and context length.

---

> > > ### Author Response · Authors · 2025-11-22
> > >
> > > 6. **The related work section omits several important studies on weighted matrix factorization, which weakens the paper’s novelty claim.**
> > >
> > >     We thank the reviewer for pointing out these related lines of work on weighted objectives and attribution-based importance measures [1, 2]. In the revised paper, we have added Section 2.3 to more thoroughly situate WSVD within this literature.
> > >
> > >     [1] Ancona, M., Ceolini, E., Öztireli, C., & Gross, M. (2017). Towards better understanding of gradient-based attribution methods for deep neural networks. arXiv preprint arXiv:1711.06104.
> > >
> > >     [2] Bach, S., Binder, A., Montavon, G., Klauschen, F., Müller, K. R., & Samek, W. (2015). On pixel-wise explanations for non-linear classifier decisions by layer-wise relevance propagation. PloS one, 10(7), e0130140.

---

> > > > ### Comment · Reviewer_YkZ1 · 2025-11-25
> > > >
> > > > Thanks for the rebuttal. With my primary concerns now addressed, I am changing my score from borderline reject to borderline accept.

---

### Official Review · Reviewer_9c8b · 2025-10-30

**Soundness:** 3
**Presentation:** 2
**Contribution:** 2
**Rating:** 6
**Confidence:** 3

**Summary:**

WSVD proposes a practical way to make vision-language models decode faster—not just smaller—by combining per-head low-rank factorization with smart tuning, quantization, and a fused kernel. It starts from the observation that plain SVD shrinks parameters and KV cache but often slows decoding because full keys/values must be reconstructed every step. WSVD instead factorizes each attention head separately so each head caches tiny latents and reconstructs only what it needs; then it uses a Fisher-weighted objective to fine-tune the low-rank factors so the most important weights are preserved. Afterward, it applies rotation-aware low-precision quantization and a short round of QAT to recover accuracy. On the systems side, a Triton kernel folds reconstruction directly into Flash Decoding so K/V are never materialized in VRAM, cutting memory traffic and kernel launches. Across LLaVA-Next and SmolVLM families on 4090/5090-class GPUs, this yields up to ~1.8× lower latency at similar cache sizes (≈25% of FP16) with ~1% average accuracy drop, and ablations show both the weighted objective and QAT are key to those results. The approach shines in memory-bound decoding (long contexts, image-heavy prompts) if brief local QAT is acceptable, though gains can vary with model and hardware.

**Strengths:**

Overall, this paper excels by sharply diagnosing why plain SVD can slow decoding and giving a compact analysis showing that per-head SVD cuts reconstruction compute and memory by a factor of r/R; it then backs that theory with a well-engineered Triton fused kernel that integrates into Flash Decoding to avoid materializing full K/V in VRAM; and it demonstrates measured gains—up to 1.8× faster decoding—while maintaining a strong accuracy–efficiency trade-off (≈1% average drop with the KV cache shrunk to ~25% of FP16). Careful ablations isolate the contributions of Fisher-weighted local fine-tuning and local QAT, and the rotation-aware quantization design (Hadamard S1 + learnable S2) is practical and lightweight compared with full end-to-end training.

**Weaknesses:**

WSVD’s evidence is somewhat narrow and system-dependent: accuracy and tuning are demonstrated mainly on ScienceQA-IMG and SEED-Bench-IMG with just ~256 calibration samples and a small set of VLMs, so generalization is unclear; its headline latency gains are reported for decoding on LLaVA-Next-7B under specific, favorable settings (e.g., batch 16, 8K tokens) on 4090/5090-class GPUs, and the paper itself notes access to high-end hardware as a limitation; practical speedups hinge on a custom Triton fused kernel integrated with Flash Decoding, raising portability questions to other runtimes/accelerators; per-head SVD increases approximation error, so Fisher-weighted fine-tuning and local QAT are required to recover accuracy—adding calibration data needs and engineering complexity; some baseline comparisons (e.g., Palu) adapt settings due to Palu’s batch-1 constraint, which complicates fairness; the method focuses on attention (Q/K/V) rather than also optimizing FFN blocks that can dominate runtime; and, until code/models are publicly released, independent replication and broader portability studies remain gated.

**Questions:**

see weakness

---

> ### Author Response · Authors · 2025-11-22
>
> We are grateful for your constructive comments. The following section summarizes your main concerns and questions, along with our detailed replies.
>
> 1. **WSVD’s generalization on models, datasets and calibration sample sizes.**
>
>     It is standard in post-training quantization and low-rank compression literature to use small calibration sets to estimate sensitive hyperparameters. For example: AWQ Section 5.3 Figure 8(a) [1] uses 8-256 samples to determine the scaling factor exponent. ASVD Section 4.1 [2] uses only 64 samples of calibration dataset in all their experiments and hyperparameter search. SVDLLM Section 4.3 [3] also uses 256 samples of calibration dataset.
>
>     Below we perform a systematic sample-size sweep from 64 to 512 on LLaVA-v1.5 and LLaVA-Next (7B) using the ScienceQA-IMG benchmark. As summarized in the following table, across both model variants averaged over 4 random seeds, the configuration using 256 samples consistently yields the highest or statistically comparable accuracy. These results indicate that our method maintains strong generalization capability even when calibrated with a limited number of samples.
>
>     | Avg acc  | | number of samples |  |  |  |
>     | --- | --- | --- | --- | --- | --- |
>     |  | $\rho_1$ | 64 | 128 | 256 | 512 |
>     | v1.5 7B | 90% | 67.52 | 67.64 | **67.83** | 67.76 |
>     | Next 7B | 90% | 69.60 | 69.57 | **69.75** | 69.62 |
>
>     To further assess generalization, the revised paper evaluates WSVD-noQ on two additional VLMs (**Appendix A.5**) and two additional benchmarks (**Appendix A.6**). Across these models and datasets, WSVD-noQ consistently matches or outperforms all baselines under comparable rank budgets, indicating that WSVD generalizes well across architectures and tasks. Note that WSVD’s decoding speedup is independent of the evaluation dataset: once the model is calibrated and compressed, runtime is determined mainly by the resulting model size and context length.
>
>     [1] Lin, Ji, et al. "Awq: Activation-aware weight quantization for on-device llm compression and acceleration." Proceedings of machine learning and systems 6 (2024): 87-100.
>
>     [2] Yuan, Zhihang, et al. "Asvd: Activation-aware singular value decomposition for compressing large language models." arXiv preprint arXiv:2312.05821 (2023).
>
>     [3] Wang, Xin, et al. "Svd-llm: Truncation-aware singular value decomposition for large language model compression." arXiv preprint arXiv:2403.07378 (2024).
>
> 2. **Latency gains on low-end GPU and under different settings.**
>
>     To address the concern about high-end hardware, the revised paper reports additional latency evaluations of LLaVA-Next-7B with WSVD-noQ on a more modest GPU, RTX 3060, under the same setting as Section 4.4. As summarized in Section 4.4 (**Speedup on Low-end GPU**), WSVD-noQ achieves up to about $2.6\times$ speedup over Flash Decoding on RTX 3060, showing that its latency benefits not only persist but become even more pronounced on low-end, bandwidth-constrained GPUs. In addition, the new **Impact of sequence length and batch size** ablation in Section 4.4 evaluates WSVD across a range of context lengths and batch sizes, and consistently shows lower latency than Flash Decoding.
>
> 3. **The custom Triton fused kernel introduces concerns about portability to other runtimes and accelerators.**
>
>     We agree that portability to other runtimes and accelerators is important. WSVD’s design does not depend on Triton. At the algorithm level, WSVD performs per-head SVD to reduce the cost of on-the-fly $K$ reconstruction during decoding (Section 3.1), and then uses lightweight local fine-tuning and QAT to recover accuracy and minimize degradation (Sections 3.2 and 3.3). At the kernel level, we accelerate decoding by streamlining on-the-fly $K$ reconstruction and attention into a single fused kernel, so that the low-rank cache is read once and immediately consumed by attention (Section 3.4). This fusion strategy is not Triton-specific: any runtime or accelerator that exposes a fused-attention (or FlashAttention-style) kernel can implement the same idea by replacing full-rank $K$ reads with low-rank reconstruction plus attention in its native programming model (e.g., CUDA/CUTLASS, TensorRT-LLM, vLLM, ROCm, or custom accelerator kernels). In our implementation we choose Triton only as a convenient vehicle for CUDA GPUs, but the underlying WSVD algorithm and fused-kernel pattern are portable to other runtimes and hardware platforms.

---

> > ### Author Response · Authors · 2025-11-22
> >
> > 4. **WSVD adds calibration data needs and engineering complexity.**
> >
> >     It is standard in post-training quantization and low-rank compression literature to use small calibration sets to estimate sensitive hyperparameters. For example: AWQ Section 5.3 Figure 8(a) [1] uses 8-256 samples to determine the scaling factor exponent. ASVD Section 4.1 [2] uses only 64 samples of calibration dataset in all their experiments and hyperparameter search. SVDLLM Section 4.3 [3] also uses 256 samples of calibration dataset. We also quantify WSVD's training cost on LLaVA-1.5-13B in **Appendix A.8** of the revised paper. Our WSVD procedure takes only **38 minutes $\approx$ 0.63 A100 GPU-hours**, whereas the official LLaVA-1.5 report states that training the 13B model requires at least **204 A100 GPU-hours**. Thus, WSVD's training cost is negligible compared to the original training cost while still providing practical decoding speedups.
> >
> >     [1] Lin, Ji, et al. "Awq: Activation-aware weight quantization for on-device llm compression and acceleration." Proceedings of machine learning and systems 6 (2024): 87-100.
> >
> >     [2] Yuan, Zhihang, et al. "Asvd: Activation-aware singular value decomposition for compressing large language models." arXiv preprint arXiv:2312.05821 (2023).
> >
> >     [3] Wang, Xin, et al. "Svd-llm: Truncation-aware singular value decomposition for large language model compression." arXiv preprint arXiv:2403.07378 (2024).
> >
> > 5. **Baseline comparisons (e.g., Palu) adapt settings due to Palu’s batch-1 constraint, which complicates fairness.**
> >
> >     To address the fairness concern around Palu’s batch-1 constraint, we further benchmark the layer-wise decoding latency of LLaVA-Next 7B on an RTX 4090 with batch size 1 for all methods. We measure the latency in FP16 for Flash Decoding, Palu, and WSVD-noQ, and set the rank ratio $\rho_2$ of both Palu and WSVD-noQ to 0.5, consistent with Section 4.4. Since the Palu paper [1] reports stronger gains at long sequence lengths, we follow that setting and evaluate long contexts from 16K to 256K tokens. Moreover, the official Palu implementation only includes the attention module, so Palu’s latency numbers here exclude the FFN module, whereas Flash Decoding and WSVD-noQ include both attention and FFN, which biases the comparison in favor of Palu. As shown in the following table, however, the trend matches our main results: even under Palu’s preferred regime, Palu remains slower than Flash Decoding, and WSVD-noQ is the fastest across all sequence lengths (e.g., at 64K tokens: Palu 3.04 ms vs. Flash 1.71 ms vs. WSVD-noQ 1.10 ms). This supports that WSVD’s latency advantages are not an artifact of using different batch sizes or settings.
> >
> >     | Seq Len | Flash Decoding | Palu | WSVD-noQ |
> >     | --- | --- | --- | --- |
> >     | 16K | 0.86 | 0.97 | 0.64 |
> >     | 32K | 1.14 | 1.67 | 0.79 |
> >     | 64K | 1.71 | 3.04 | 1.1 |
> >     | 128K | 2.85 | 5.8 | 1.7 |
> >     | 256K | 5.13 | 11.32 | 2.89 |
> >
> >     [1] Chang, Chi-Chih, et al. "Palu: Compressing kv-cache with low-rank projection." arXiv preprint arXiv:2407.21118 (2024).

---

> > > ### Author Response · Authors · 2025-11-22
> > >
> > > 6. **The method focuses on attention (Q/K/V) rather than also optimizing FFN blocks that can dominate runtime.**
> > >
> > >     Prior profiling studies of large language models [1–4] show that self-attention layers, especially the key and value (KV) cache, account for a substantial portion of decoding latency, energy, and memory usage, and often become the primary bottleneck for inference speed. WSVD therefore focuses on attention: we apply per-head SVD to Q/K/V to shrink the KV cache and reduce the cost of on-the-fly reconstruction. As detailed in Section 4.1, we also optimize FFN blocks via post-training quantization, and quantize all FFN layers in our models, which already provides significant computational savings. Our approach is furthermore compatible with other FFN-optimization techniques, and these can be combined with WSVD in future work.
> > >
> > >     Moreover, VLMs introduce hundreds to thousands of image tokens per example, which further increases the decoding context length and makes KV-cache I/O an even more severe bottleneck than in text-only LLMs. By substantially shrinking the KV cache via WSVD and fusing low-rank reconstruction with attention, our method directly alleviates this bottleneck and thus translates these algorithmic changes into practical speedups, as detailed in Section 4.4.
> > >
> > >     [1] Hooper, Coleman, et al. "Kvquant: Towards 10 million context length llm inference with kv cache quantization." Advances in Neural Information Processing Systems 37 (2024): 1270-1303.
> > >
> > >     [2] Kim, Minsu, et al. "Oaken: Fast and Efficient LLM Serving with Online-Offline Hybrid KV Cache Quantization." Proceedings of the 52nd Annual International Symposium on Computer Architecture. 2025.
> > >
> > >     [3] Kwon, Woosuk, et al. "Efficient memory management for large language model serving with pagedattention." Proceedings of the 29th symposium on operating systems principles. 2023.
> > >
> > >     [4] Zhang, Zhenyu, et al. "H2o: Heavy-hitter oracle for efficient generative inference of large language models." Advances in Neural Information Processing Systems 36 (2023): 34661-34710.
> > >
> > > 7. **Until code/models are publicly released, independent replication and broader portability studies remain gated.**
> > >
> > >     We agree that public release is essential for independent replication and portability studies. Upon acceptance, we will open-source the complete WSVD codebase, including preprocessing scripts, calibration and tuning pipelines, model checkpoints, and fused decoding kernels. This release will enable the community to fully reproduce our results and to adapt WSVD to additional runtimes and hardware accelerators.

---

> > > > ### Author Response · Authors · 2025-11-27
> > > >
> > > > Dear reviewer 9c8b, please let us know if our response has addressed your questions. Thank you.

---

### Official Review · Reviewer_9Jd6 · 2025-10-31

**Soundness:** 3
**Presentation:** 3
**Contribution:** 3
**Rating:** 6
**Confidence:** 2

**Summary:**

This paper introduces WSVD, a weighted low-rank approximation framework for efficient low-precision vision-language models (VLMs). It addresses latency issues of existing SVD methods by applying fine-grained per-head SVD, assigning weight importance via Fisher information to preserve accuracy, and integrating quantization-aware training. A fused kernel merges low-rank reconstruction with Flash Decoding to cut I/O overhead. Experiments on 5 VLMs show WSVD achieves up to 1.8× decoding speedup, maintains accuracy (≤1% drop even at 50% parameter compression), and reduces KV cache to 25% of FP16 models, outperforming baselines like ASVD and SVD-LLM

**Strengths:**

1. In terms of originality, the paper innovatively combines fine-grained per-head SVD, Fisher information-based weight importance scoring, and quantization-aware training with a fused kernel design, effectively addressing the latency-accuracy trade-off in VLM optimization that prior SVD methods failed to resolve.
2. Regarding quality, the work demonstrates rigorous experimental validation across 5 VLMs, 2 benchmarks, and multiple GPUs, with detailed ablation studies confirming the effectiveness of each component and fair comparisons against diverse baselines.
3. For significance, the WSVD framework achieves up to 1.8× decoding speedup while preserving accuracy and reducing KV cache to 25% of FP16 models, providing a practical solution for efficient VLM deployment on resource-constrained devices.

**Weaknesses:**

1. The paper’s evaluation of WSVD’s generalization across VLM architectures is limited, as it only tests models from the LLaVA family and SmolVLM—excluding other representative VLMs like BLIP-2 or Qwen-VL. This gap makes it unclear if WSVD’s per-head SVD and fused kernel design work equally well for VLMs with different vision-text fusion mechanisms (e.g., Qwen-VL’s high-resolution perception), and expanding tests to these models would strengthen its generalizability.
2. The analysis of WSVD’s performance under varying sequence lengths is insufficient; experiments only use a fixed sequence length of 8192, while real-world VLM use cases (e.g., long image captions or multi-image QA) involve variable lengths. Without testing how WSVD’s latency and accuracy scale with shorter/longer sequences, it is hard to assess its adaptability to diverse input scenarios, which is critical for practical deployment.
3. The paper lacks a detailed comparison of WSVD’s computational overhead during the fine-tuning phase. While it mentions local fine-tuning (100 steps) and QAT (50 steps) are lightweight, it does not quantify this overhead (e.g., training time, GPU memory usage) against end-to-end finetuning or baselines like QSVD’s optimization process. Providing such data would help users evaluate if WSVD’s efficiency gains justify its fine-tuning costs, especially for resource-limited teams.

**Questions:**

1. Could you provide experimental results of WSVD on other representative VLMs (e.g., BLIP-2, Qwen-VL) beyond the LLaVA family and SmolVLM? This would clarify whether WSVD’s design adapts to VLMs with different vision-text fusion mechanisms .
2. Can you supplement latency and accuracy data of WSVD under variable sequence lengths (not just 8192)? This helps assess its adaptability to real-world scenarios with short/long inputs .
3. Could you quantify the computational overhead (e.g., training time, GPU memory) of WSVD’s local fine-tuning and QAT, and compare it with baselines like QSVD? This enables evaluating if its efficiency gains justify the tuning costs .

---

> ### Author Response · Authors · 2025-11-22
>
> We are grateful for your constructive comments. The following section summarizes your main concerns and questions, along with our detailed replies.
>
> 1. **Experimental results of WSVD on other representative VLMs (e.g., BLIP-2, Qwen-VL) beyond the LLaVA family and SmolVLM.**
>
>     Thank you for the suggestion. To further examine the generality of WSVD beyond the LLaVA family and SmolVLM, we additionally apply WSVD to Qwen-VL-7B and Molmo-7B-O, following the same evaluation setting as in Table 1. The new results are reported in the revised paper in **Appendix A.5**, and indicate that WSVD transfers well to VLMs with different vision–language fusion designs, supporting the general applicability of our method.
> 2. **Supplement latency and accuracy data of WSVD under variable sequence lengths.**
>
>     Thank you for the suggestion. In the revised paper, we add a decoding latency ablation over sequence length and batch size for LLaVA-Next 7B under the same setting as Section 4.4, and report the results in Section 4.4 (**Impact of Sequence Length and Batch Size**) and **Tables 5 and 6**. Across all tested sequence lengths and batch sizes, WSVD-noQ still achieves substantial decoding speedups over FP16 Flash Decoding. As for accuracy, Section 4 and Appendix A.6 in the revised paper evaluate WSVD on a diverse set of benchmarks whose effective input lengths are determined by each test sample (from short QA-style prompts to longer multimodal instructions), and across all of these tasks WSVD (with and without quantization) shows only minimal degradation compared to the FP16 baselines.
>
> 3. **Quantify the computational overhead (e.g., training time, GPU memory) of WSVD’s local fine-tuning and QAT, and compare it with baselines like QSVD.**
>
>    We quantify the computational overhead of WSVD and compare it with QSVD on LLaVA-1.5 13B, and report the detailed results in **Appendix A.8**. As summarized there, WSVD’s overall calibration takes only 38 minutes (≈0.63 A100 GPU-hours), and both the local FT and QAT stages fit within a peak memory footprint of 15 GB, which is about 2.5× less tuning time than QSVD and negligible compared to the 204 A100 GPU-hours needed to train LLaVA-1.5-13B from scratch.

---

> > ### Comment · Reviewer_9Jd6 · 2025-11-26
> >
> > Thank you for the response.I am pleased with the clarifications and I will maintain my positive score to recommend acceptance.

---

### Official Review · Reviewer_ewCT · 2025-11-02

**Soundness:** 2
**Presentation:** 3
**Contribution:** 2
**Rating:** 6
**Confidence:** 3

**Summary:**

This paper proposes WSVD. By leveraging SVD, local weighted fine-tuning, and quantization-aware training, the method addresses the issues of high latency and accuracy loss in low-precision quantization of VLMs. Its fine-grained SVD reduces memory access overhead, and weighted decomposition preserves the importance of critical weight. The authors claim that the method achieves a 1.8× speedup in VLM decoding.

**Strengths:**

This paper adopts per-head SVD operations, reducing memory overhead and latency in KV cache reconstruction. The method has a clear motivation, targeting the high memory access latency of traditional SVD. The paper has clear writing logic and readability, with a rigorous structure.

**Weaknesses:**

The paper lacks sufficient analysis on the theoretical basis for per-head SVD rank selection, and the sensitivity validation of weighted scoring is absent. Baseline comparisons fail to include some of the latest low-rank quantization methods, and the performance stability in extreme low-rank scenarios has not been fully explored.

**Questions:**

I mainly have some questions about the validation of the method.
1. I am quite curious about the basis for selecting rank r in per-head SVD. Is there any quantitative analysis on the specific impacts of different rank settings on model performance and latency?
2. When using the Fisher information score as the weight importance indicator in weighted fine-tuning, has it been compared with other indicators (such as gradient magnitude)? What is the proportion of its contribution to performance improvement?
3. In quantization-aware training, what impact do the settings of orthogonal matrices S₁ and S₂ have on the quantization effect?

---

> ### Author Response · Authors · 2025-11-22
>
> We are grateful for your constructive comments. The following section summarizes your main concerns and questions, along with our detailed replies.
>
> 1. **Impacts of different rank settings on model performance and latency, and the performance stability in extreme low-rank scenarios.**
>
>     Our FP16 results in paper Table 1 sweep a wide range of parameter ratios $\rho_1$ (from $0.9$ down to $0.5$) for WSVD-noQ and all SVD baselines, directly showing how different rank budgets affect accuracy: as $\rho_1$ and the corresponding $\rho_2$ decrease, accuracy degrades gradually.
>
>     To further quantify the impact of rank in the WSVD quantized setting and to probe more extreme low-rank regimes, we additionally sweep $\rho_1$ from $0.9$ down to $0.3$ for LLaVA-v1.5 13B and LLaVA-Next 13B using the same QAT and quantization process as in paper Table 2, with ScienceQA as the evaluation benchmark. As summarized in the following Table, WSVD remains very close to the FP16 baseline for $\rho_1 \in [0.9, 0.5]$ (and even slightly exceeds it at some settings), and only shows a larger drop when pushing to $\rho_1 = 0.3$, which corresponds to an extreme low-rank configuration.
>
>     | **ScienceQA-IMG** | Method | $\rho_1$ | $\rho_1$ | $\rho_1$ | $\rho_1$ | $\rho_1$ | $\rho_1$ | $\rho_1$ |
>     | --- | --- | --- | --- | --- | --- | --- | --- | --- |
>     | | |90% | 80% | 70% | 60% | 50% | 40% | 30% |
>     | LLaVA-v1.5 13B | FP16 | 71.83 |  |  |  |  |  |  |
>     |  | WSVD | 71.19 | 71.39 | 71.49 | 72.04 | 72.14 | 69.16 | 63.41 |
>     | LLaVA-Next 13B | FP16 | 73.23 |  |  |  |  |  |  |
>     |  | WSVD | 72.48 | 72.09 | 72.38 | 73.33 | 73.08 | 71.49 | 62.87 |
>
>     The relationship between rank and decoding latency is further summarized in the table below and in the revised paper (Section 4.4, **Impact of Rank Ratio**). Using the same setup as Section 4.4, we vary the rank ratio $\rho_2 \in \lbrace0.9, 0.7, 0.5\rbrace$ for WSVD-noQ and measure latency (in ms) on RTX 4090 and RTX 5090 GPUs. The results show that smaller $\rho_2$ (lower ranks) consistently reduce decoding latency, confirming that our fused kernel translates lower ranks into practical speedups.
>
>     | GPU | Flash Decoding |$\rho_2$  | $\rho_2$ | $\rho_2$ |
>     | --- | --- | --- | --- | --- |
>     |  |  | 90% | 70% | 50% |
>     | 4090 | 2.92 | 2.83 | 2.53 | 1.66 |
>     | 5090 | 2.14 | 1.87 | 1.75 | 1.15 |
> 2. **Comparisons with other weight importance indicators (such as gradient magnitude).**
>
>     We thank the reviewer for the insightful question. We experimented with several types of importance scores and found that the Fisher information–based score performs the best among the alternatives.
>
>     To compare Fisher information with other weight importance indicators, we conducted an ablation on LLaVA-1.5 7B using the same calibration set as in the main paper. Specifically, we replaced the Fisher-based weight score with the average gradient magnitude on the calibration set (the mean gradient magnitude of each weight) and used these gradient-based scores as the weighting coefficients in our weighted fine-tuning. We evaluate the local fine-tuned model on the ScienceQA test set and sweep the parameter ratio $\rho_1 \in [0.9, 0.5]$.
>
>     The results are summarized in the following table. Across all compression settings, Fisher-based weighting consistently outperforms gradient-magnitude weighting, shows the superiority of the Fisher-based approaches. This is because, as the LLM approaches convergence, most gradients become close to zero, making gradient magnitude an unreliable indicator.
>
>     | $\rho_1$ | Grad magnitude | Fisher-based |
>     | --- | --- | --- |
>     | 90% | 66.14 | **68.17** |
>     | 80% | 66.78 | **67.72** |
>     | 70% | 64.80 | **67.28** |
>     | 60% | 63.96 | **65.89** |
>     | 50% | 63.36 | **65.49** |

---

> ### Author Response · Authors · 2025-11-22
>
> 3. **Impact of orthogonal matrices S₁ and S₂ on the quantization-aware training effect.**
>
>     To isolate the impact of the orthogonal matrices $\mathbf{S}_1$ and $\mathbf{S}_2$ on quantization-aware training, we conduct an ablation where we remove $\mathbf{S}_1,\mathbf{S}_2$ during local QAT while keeping all other settings identical to Table 2. Concretely, the “QAT w/o $\mathbf{S}_1,\mathbf{S}_2$” variant performs local QAT directly on the low-rank factors without the additional orthogonal transforms.
>     We evaluate on ScienceQA-IMG for four models. As summarized in the following table, WSVD consistently outperforms the variant without $\mathbf{S}_1,\mathbf{S}_2$ on all models, with an average gain of about $0.9$ accuracy points, indicating that the orthogonal matrices have a positive impact on QAT.
>
>     | Method | Science QA-IMG |  |  |  |  |
>     | --- | --- | --- | --- | --- | --- |
>     |  | Llava-v1.5 7B | Llava-v1.5 13B | Llava-Next 7B | Llava-Next 13B | Average |
>     | QAT w/o $\mathbf{S}_1,\mathbf{S}_2$ | 63.41 | 71.24 | 66.09 | 72.18 | 68.23 |
>     | WSVD | 64.25 | 72.14 | 66.94 | 73.08 | 69.10 |
>
>     This improvement can be attributed to the additional orthogonal reparameterization introduced by $\mathbf{S}_1$ and $\mathbf{S}_2$, which suppresses channel-wise outliers and better balances the dynamic range across channels, leading to a better-conditioned initialization for local QAT. In addition, the learnable $\mathbf{S}_2$ increases the flexibility of QAT in adapting to quantization noise. Together, these effects enhance the model’s ability to absorb quantization error compared to QAT without these orthogonal transforms.
>
> 4. **Baseline comparisons with some of the latest low-rank quantization methods.**
>
>     To the best of our knowledge, QSVD is the only prior work that explicitly combines low-rank decomposition with quantization and applies quantization to both weights and activations, so we include QSVD as a main baseline (Table 2). Other low-rank methods such as ASVD and SVDLLM are primarily FP16 compression techniques and do not jointly optimize low-rank structure and quantization. We therefore use them as FP16 baselines and additionally introduce QASVD as a quantized variant of ASVD. Palu instead focuses on compressing and quantizing the KV cache via group-head SVD rather than quantizing all model weights and activations. In the main paper, we compare Palu and WSVD-noQ in system-level FP16 decoding latency (Section 4.4). Consistent with the settings in Section 4.2, we further provide an accuracy comparison under matched compression and whole-model W8A8 quantization configurations for Palu and WSVD, rather than quantizing only the KV cache for Palu, to ensure fairness. As shown in the following table, WSVD improves ScienceQA accuracy over Palu and further narrows the gap to the FP16 models.
>     | Method | Science QA-IMG | | | | |
>     | ------ | -------------- | -------------- | ------------ |-------------- |------------|
>     |        | Llava-v1.5 7B  | Llava-v1.5 13B | Llava-Next 7B | Llava-Next 13B | Avg |
>     | Palu   | 65.01          | 70.93 | 65.73 | 71.59 | 68.32 |
>     | WSVD  | 64.25          | 72.14 | 66.94 | 73.08 | 69.10 |
>     | FP16   | 68.10          | 71.83 | 69.60 | 73.23 | 70.69 |

---

> > ### Author Response · Authors · 2025-11-27
> >
> > Dear reviewer ewCT, please let us know if our response has addressed your questions. Thank you.

---

> > > ### Comment · Reviewer_ewCT · 2025-11-28
> > >
> > > Thank you for your response. I will maintain my evaluation score.

---

### Author Response · Authors · 2025-12-03
**Summary Comment**

We are grateful to the reviewers for their thoughtful and supportive feedback. Collectively, the four reviews consistently highlight several core strengths of our work:

- **Clear motivation and practicality:** WSVD is viewed as a practical and well-motivated framework that directly addresses the high memory-access latency and decoding slowdown issues of traditional SVD.
- **Per-head SVD and kernel design:** Reviewers highlighted our fine-grained per-head SVD with clear compute–memory overhead analysis, and our fused kernel integrated into Flash Decoding, which reduces KV cache reconstruction overhead without materializing full K/V in VRAM.
- **Comprehensive compression framework:** The method is considered original for combining per-head SVD, Fisher-based importance weighting, local fine-tuning, and QAT within a system that jointly leverages low-rank decomposition and quantization.
- **Strong empirical validation:** Reviewers praised the rigorous experiments across 5 VLMs, 2 benchmarks, and multiple GPUs, along with careful ablations and fair comparisons to diverse baselines.
- **Practical efficiency gains:** WSVD achieves up to 1.8× decoding speedup and reduces the KV cache to about 25% of FP16 with only $\approx$ 1% average accuracy drop, making it practical for resource-constrained devices.
- **Clarity of presentation:** Reviewers consistently noted the paper's clear writing, logical organization, and readability.

We appreciate the reviewers’ positive assessments. In response to their comments, we have incorporated several revisions and clarifications, while all main conclusions of the paper remain unchanged:

1. **Expanded Empirical Validation Across Models and Benchmarks**
    To strengthen claims of generality, we substantially expanded our evaluation:
    - **Broader model coverage:** Added experiments on **Qwen-VL 7B** and **Molmo-7B-O** in the revised paper, where WSVD-noQ consistently surpasses ASVD and SVDLLM at most rank ratios and often matches or slightly exceeds FP16 accuracy.
    - **More diverse benchmarks:** Added **HRBench-4K**, and **OCRBench** in the revised paper, all evaluated using the same 256-sample calibration set. WSVD-noQ maintains the strongest or comparable accuracy across nearly all settings.
    - **Calibration robustness:** Conducted a sample-size sweep (64 → 512 samples). The 256-sample configuration achieves the best or tied-best performance, demonstrating stability even with limited calibration data.

2. **Additional Latency Analyses Across Sequence Lengths, Batch Sizes, and Hardware**
    To address system-level concerns, we expanded our latency studies:
    - **Sequence length scaling:** For LLaVA-Next-7B, WSVD-noQ provides **1.3×–1.8×** speedups on RTX 4090 and up to **2.35×** on RTX 5090 when sequence length grows from 1K → 32K tokens.
    - **Batch size effects:** At 8K tokens, speedups range **1.4×–1.9×** as batch size increases from 4 → 64.
    - **Lower-end hardware:** On **RTX 3060**, WSVD-noQ shows even larger improvements (e.g., **2.59×** at 16K tokens), confirming amplified gains on memory-bandwidth-limited devices.
    - **Palu comparison under batch size 1:** Even under Palu’s preferred long-context, batch size 1 setting, WSVD-noQ remains the fastest (e.g., at 64K tokens: **1.10 ms vs. 1.71 ms vs. 3.04 ms**).

3. **Methodological Clarifications on SVD Formulation and Fisher-Based Weighting**
    We provided detailed clarifications on key methodological aspects:
    - **Per-head vs. full-matrix SVD:** Matching relative rank implies $R = N_{\text{heads}} \cdot r$, giving WSVD a **1/N_heads** reconstruction cost (32×–40× reduction in our models).
    - **Weighted objective formulation:** Explained why $F^{1/2}$ naturally arises from the Fisher-weighted second-order approximation.
    - **WSVD vs. FWSVD:** FWSVD uses a **row-wise Fisher pre-scaling**, whereas WSVD applies **element-wise Fisher** during both fine-tuning and QAT, providing finer-grained control.

4. **Ablations on Importance Metrics**
    We added extensive comparisons to justify the use of Fisher information as the importance signal:
    - Compared Fisher to gradient magnitude, integrated gradients, and input×grad. Fisher consistently achieves the strongest results (e.g., at ratio 70% on LLaVA-1.5-7B: **67.28 vs. 64.80 vs. 45.56 vs. 45.12**).
    - These results show Fisher remains informative even when gradients vanish and outperforms attribution-based metrics when repurposed for parameter importance.

---

> ### Author Response · Authors · 2025-12-03
>
> 5. **Ablations on QAT Components (Orthogonal Transforms $S_1, S_2$)**
>     To isolate the contributions of orthogonal reparameterization:
>     - Removing $S_1, S_2$ yields an average **~0.9% accuracy drop** across four models.
>     - The transforms suppress channel-wise outliers and balance activation ranges, improving QAT stability and resilience to quantization noise.
>
> 6. **Quantified Tuning Cost and Efficiency Advantages**
>     We benchmarked tuning overhead against QSVD under matched conditions:
>     - **WSVD:** 38 minutes (0.63 A100-hours), **15 GB** peak memory for local FT and QAT stages.
>     - **QSVD:** 96 minutes (1.6 A100-hours).
>     - WSVD is thus **≈2.5× more efficient**, and tuning cost is negligible relative to the **204 A100-hours** required for training LLaVA-1.5-13B.
>
> 7. **Fairer Baseline Comparisons and Revised Related Work**
>     To improve fairness and completeness:
>     - **Matched-quantization comparison with PaLU:** Under whole-model W8A8, WSVD achieves higher accuracy (e.g., **72.14 vs. 70.93** on LLaVA-1.5-13B).
>     - **Expanded related work:** Added Fisher-based pruning, weighted low-rank approximation, and attribution-based relevance literature to better situate WSVD’s novelty as an element-wise Fisher-guided compression framework.
>
> 8. **Portability and Runtime Deployment**
>     To address concerns about portability:
>     - Clarified that the fused decoding kernel is **not tied to Triton** and can be implemented in CUDA/CUTLASS, TensorRT-LLM, vLLM, ROCm, or accelerator-specific kernels.
>     - Upon acceptance, we will release the full WSVD codebase, including calibration, tuning, and fused kernel implementations.
>
> After our revision and clarification, the final scores are **6**, **6**, **6**, and **6**: **all reviewers maintained positive evaluations**, and **one reviewer raised the score from borderline reject to borderline accept**, resulting in a stronger and unanimous recommendation for acceptance.

---

### Meta-Review · Area_Chair_s1P5 · 2026-01-03

**Summary:**

The authors made a successful rebuttal, and the review score rose from 4666 to 6666. After reading the paper and discussions, the AC tends to accept this paper.

**Reviewer Concerns:**

Most concerns are solved by the rebuttal

**Reviewer Scores:**

According to the discussion, three of the reviewers are convinced by the rebuttal and give a positive score to the paper.
The reviewer 9c8b did not participate in the discussion, while the initial review score is positive.

---

### Decision · Program_Chairs · 2026-01-26

Accept (Poster)